# Maintenance of cytoplasmic and membrane densities shapes cellular geometry in *Escherichia coli*

**Griffin Chure** [1] ✉, **Roshali T. de Silva**[1,4], **Richa Sharma**[1], **Michael C. Lanz**[1,2] & **Jonas Cremer**[1,3] ✉

Microbes precisely control their composition and geometry across diverse growth conditions, yet the mechanisms coordinating these processes remain unclear. Here, we integrate quantitative proteomics, microscopy, and biochemical measurements to reveal a biophysical principle linking these properties in *Escherichia coli*: cytoplasmic and membrane protein densities maintain a tightly conserved ratio across growth conditions, while the periplasmic density varies. Building on this observation, we develop a mathematical model demonstrating that maintaining this density ratio constrains the surface-to-volume ratio as a nonlinear function of proteome composition, specifically the ribosomal proteome fraction and partitioning between cellular compartments. The model holds under guanosine tetraphosphate perturbations that alter ribosome levels, further demonstrating that cellular geometry is not strictly determined by growth rate. These findings provide a biophysical framework for geometry control, underscoring density maintenance as a key physiological constraint that shapes cellular phenotypes.

Microbial cells demonstrate a remarkable phenotypic plasticity, simultaneously regulating their size as well as their macromolecular composition in concert with their growth rate across diverse conditions[1]. Research on this plasticity has largely proceeded along separate lines, resulting in a set of phenomenological "growth laws" that independently describe how cell composition[2–14] and cell geometry[15–26] scale with growth rate. This has been most prominently studied in the model bacterium *Escherichia coli*, where the cellular ribosome content (Supplementary Fig. 1a) and cell volume (Supplementary Fig. 1b) scale approximately linearly and exponentially with the steady-state growth rate, respectively. Inspired by these strong phenomenological relations, several studies have interrogated their plausible interconnection[27–29], yet we lack a unifying framework that directly relates compositional and geometric regulation across diverse environments.

In this work, we integrate a systematic experimental dissection of compositional and geometric covariation in *E. coli* across growth conditions with biophysically-grounded mathematical modeling to derive a predictive view of how proteomic composition and partitioning between cellular compartments jointly influence cellular geometry. Using quantitative mass spectrometry, we reveal a strong growth-rate dependence in how *E. coli* partitions its proteome between the cytoplasm, periplasm, and cell membranes. Specifically, we show that the cytoplasmic and periplasmic proteome partitions are strongly anti-correlated, suggesting a trade-off in localization between these compartments, while the membrane fraction of the proteome is constant across growth conditions. Given these relations, we propose that the surface-to-volume ratio is controlled such that the macromolecular densities within these compartments are tightly constrained. We present a biophysical model centered on this density-maintenance hypothesis that quantitatively describes how the surface-to-volume ratio is directly dependent on the cellular composition, thereby providing a link between the phenomenological growth laws.

[1]Department of Biology, Stanford University, Stanford, CA, USA. [2]Chan Zuckerberg Biohub, San Francisco, CA, USA. [3]Bio-X, Stanford University, Stanford, CA, USA. [4]Present address: School of Life Sciences, Arizona State University, Tempe, AZ, USA. ✉e-mail: griffinchure@gmail.com; jbcremer@stanford.edu

## Results

### Condition-dependent proteome composition

Interrogating the relationship between compositional and size regulation demands a self-consistent dataset where macromolecular composition, proteomic localization, cell geometry, and growth rates are simultaneously measured across different environmental conditions. Using the Gram-negative bacterium *E. coli* as a model system, we conducted a comprehensive suite of experiments to simultaneously measure these quantities in steady-state across seven different growth conditions defined by different carbon sources or mixes of carbon sources (described in Methods), yielding a data set highly-consistent with aggregated data from the literature (Supplementary Fig. 2). Leveraging recent advancements in quantitative mass spectrometry[30] and localization annotation of the *E. coli* proteome[31], we queried how the *E. coli* proteome is partitioned between cellular compartments, namely the cytoplasm, periplasm, and membranes (Fig. 1a).

The fraction of the proteome partitioned to each compartment exhibits distinct scaling relationships with growth rate. The cytoplasmic proteome partition ($\psi_{cyto}$) increases approximately linearly as a function of the growth rate, ranging from ~75 to 85% of the total proteome (Fig. 1b). In contrast, the periplasmic partition ($\psi_{peri}$) decreases with increasing growth rate, ranging from ~15 to 5% (Fig. 1c). The partitioning of membrane-associated proteins ($\psi_{mem}$) remains approximately constant at ~12%, exhibiting no significant dependence on growth rate (Fig. 1d). Notably, the cytoplasmic and periplasmic partitioning trends are near-perfectly anticorrelated across growth rates (Fig. 1e), in line with a constant membrane partition $\psi_{mem}$. This is further illustrated by a constant total partition of the proteome to the cytoplasm and periplasm ($\psi_{cyto} + \psi_{peri}$) across all growth conditions, accounting for ~87% of the proteome mass (Fig. 1f). We further performed a systematic comparative analysis of proteomics datasets from the literature, which, while noisier, confirmed the described trend (Supplementary Fig. 4a–g). In sum, these observations suggest a strong trade-off in how cells structure their proteome partitioning: any

increase in cytoplasmic protein load is offset by a decrease in periplasmic protein load and vice versa, while the membrane protein partitioning remains unchanged. This is consistent with theoretical predictions that metabolic strategies requiring expanded periplasmic allocation come at the expense of cytoplasmic biosynthetic capacity[32].

### Conserved density balance between membrane and cytoplasm

Trade-offs in the functional composition of the proteome have been well studied in *E. coli*, particularly the competitive synthesis of ribosomal and metabolic proteins[3,7,33,34] to promote biosynthetic fluxes and growth[11,35,36]. We propose that the trends shown in Fig. 1 illustrate that proteome partitioning across cellular compartments is also governed by a resource allocation constraint, but with a distinctly different underlying biophysical principle: In order to maintain macromolecular densities, which are critical in controlling biochemical reaction rates[37,38] and are therefore subject to evolutionary optimization[39–41], changes in proteome partitioning must be accompanied by compensatory changes in compartment size.

Using our direct measurements of cell size in addition to proteomic and bulk biochemical measurements (see Methods), we examined how, if at all, the macromolecular masses (Supplementary Fig. 5) and densities (Fig. 2) within each compartment change across conditions. The cytoplasmic density $\rho_{cyto}$, comprised primarily by the total protein and RNA mass per unit cytoplasmic volume, remains moderately stable across conditions at approximately 400 fg/$\mu$m$^3$, with a weak linear dependence on growth rate (Fig. 2a). Similarly, we find that the areal density of proteins within the membranes $\sigma_{mem}$ is tightly constrained at approximately 3 fg/$\mu$m$^2$, with minimal variation across conditions (Fig. 2b). In contrast, the periplasmic protein density exhibits a markedly different scaling behavior, decreasing from approximately 200–50 fg/$\mu$m$^3$ as a function of growth rate (Fig. 2c). Despite this steep decline in periplasmic density, the total mass of periplasmic proteins per cell remains constant at approximately 20 fg for all growth conditions (Supplementary Fig. 5e), suggesting that

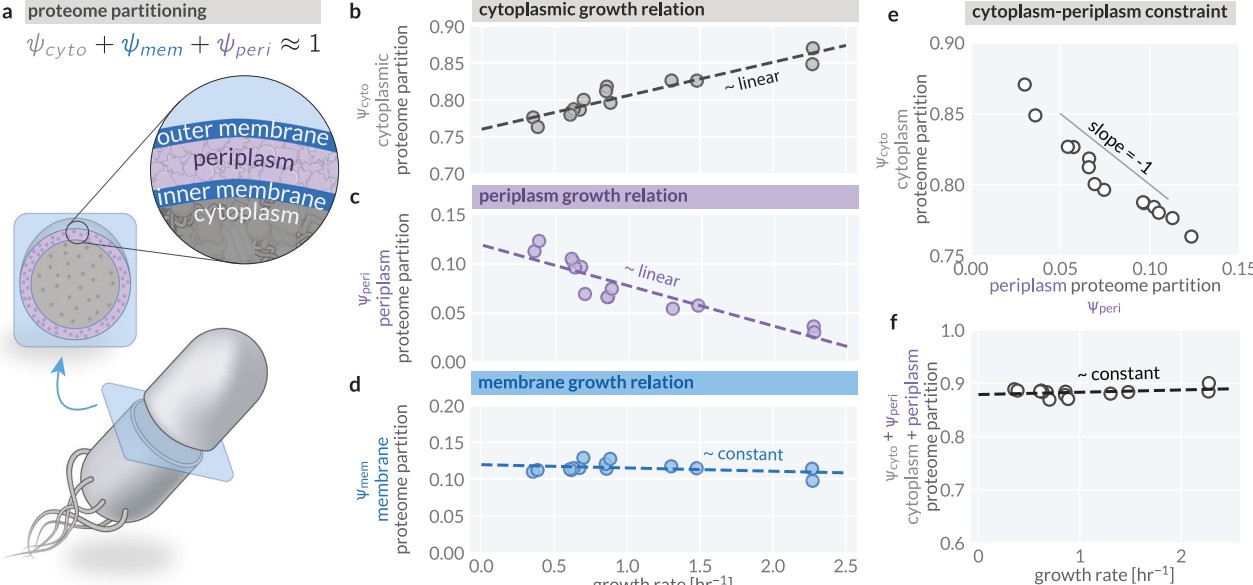

**Fig. 1 | Proteomic partitioning and growth relations across *E. coli*'s cellular compartments. a** *E. coli*'s three distinct, spatially separated compartments—the cytoplasm (gray), membranes (blue), and intermembrane space termed the periplasm (purple). The proteome is partitioned between these compartments. The mass fraction of the total proteome occupied by **b** the cytoplasm $\psi_{cyto}$, **c** the periplasm $\psi_{peri}$, and **d** inner and outer membranes $\psi_{mem}$ as a function of the steady-state growth rate. Dashed lines represent empirical linear regressions on the data, with slopes as follows: $\psi_{cyto} \rightarrow \approx 0.05$ h; $\psi_{peri} \rightarrow \approx -0.04$ h; $\psi_{mem} \rightarrow \approx -0.004$ h. **e** The

linear trade-off correlation between the proteome allocation toward the cytoplasm and periplasm with a slope of $\approx -1$. **f** The sum total proteome mass fraction of the cytoplasm and periplasm $\psi_{cyto} + \psi_{peri}$ as a function of the growth rate. Dashed line is a linear regression on the data with a slope of $\approx 0.004$ h. See Supplementary Table 1 for values of slopes, intercepts, and their normalized values. See also Supplementary Fig. 4 for a comparative analysis of proteome partitioning across literature datasets.

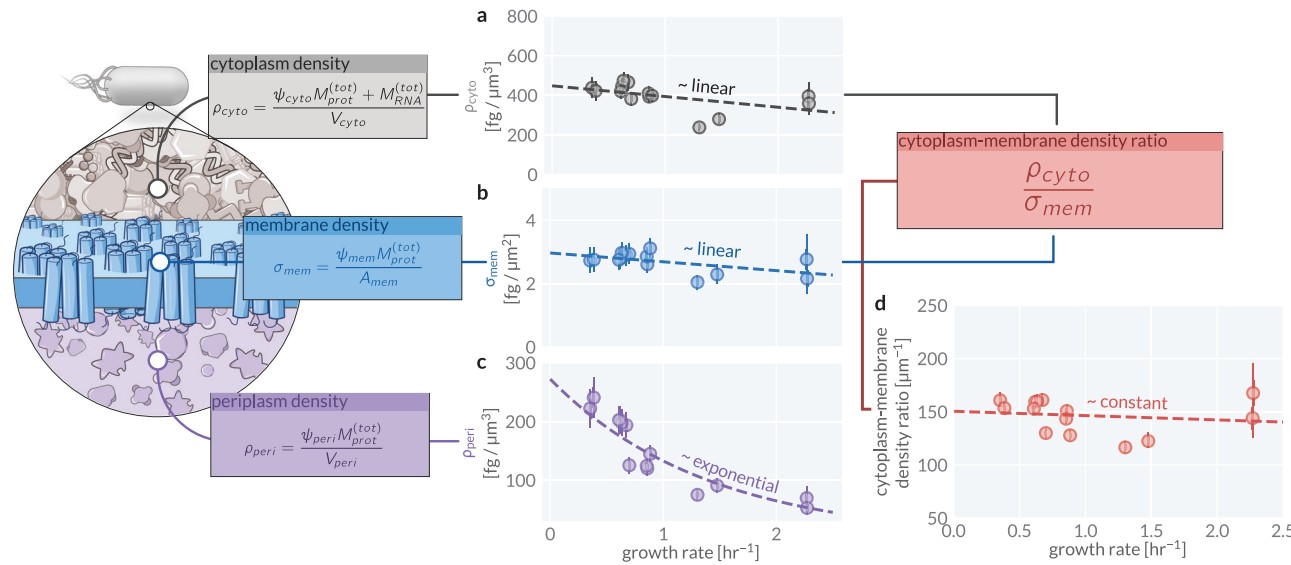

**Fig. 2 | Macromolecular densities within cellular compartments are tightly regulated.** The major macromolecular densities as a function of the growth rate in (**a**) the cytoplasm, **b** the membranes, and **c** the periplasmic space. **d** The ratio of the cytoplasmic and membrane densities as a function of the growth rate. Points calculated from mass spectrometry data as described in boxes. Total protein mass $M_{prot}^{(tot)}$ and total RNA mass $M_{RNA}^{(tot)}$ per cell for each sample was empirically determined from independent quantification as described in the Supplementary Note 3. Points represent the mean value of the posterior probability distribution for each quantity. Extent of the thin and thick error bars correspond to the 95% (approximately $2\sigma$) and 68% (approximately $1\sigma$) credible regions of the posterior distribution, respectively. Dashed lines represent fits to the mean values. See also Supplementary Fig. 4 for a comparative analysis of densities using different data compiled from the literature.

periplasmic protein load, rather than the density, is homeostatically maintained.

While both the cytoplasmic density and membrane protein density exhibit slight negative linear dependencies on growth rate, their ratio remains constant across conditions (Fig. 2d). We also observe similar density trends in our comparative analysis of literature datasets (Supplementary Fig. 4). While additional assumptions were required to infer densities from these data, the results further support a well-constrained range of density ratios across most datasets. Given the prevalence of interfacial biochemical interactions between the cytoplasm and the membranes, such a constant density ratio is likely physiologically important with evolutionary pressure to maintain this density ratio influencing cell-size control.

### Deriving a model of density maintenance

To better understand the constraints that a constant cytoplasm-membrane density imposes on cell geometry, we can mathematically examine how each density is defined. We make the well-motivated assumption that the majority of the cytoplasmic biomass is composed of protein and RNA[42], yielding the cytoplasmic density

$$\rho_{cyto} = \frac{\psi_{cyto} M_{prot}^{(tot)} + M_{RNA}^{(tot)}}{V_{cyto}}, \tag{1}$$

where $\psi_{cyto}$ represents the cytoplasmic proteome partition, $M_{prot}^{(tot)}$ is the total protein mass per cell, $M_{RNA}^{(tot)}$ is the total RNA mass per cell, and $V_{cyto}$ is the total cytoplasmic cell volume, defined as the total cellular volume $V$ less the periplasmic volume, $V_{cyto} = V - V_{peri}$. Similarly, we define the areal density of proteins within the cell membranes $\sigma_{mem}$ as

$$\sigma_{mem} = \frac{\psi_{mem} M_{prot}^{(tot)}}{A_{mem}}, \tag{2}$$

where $\psi_{mem}$ denotes the membrane proteome partition and $A_{mem}$ is the total membrane area of the cell, including both inner and outer membranes.

We now introduce two geometrically well-justified simplifications. First, we consider that the total cytoplasmic volume $V_{cyto}$ is sufficiently larger than the total periplasmic volume $V_{peri}$, such that $V_{cyto} \approx V$. Second, we assume that the inner and outer membranes are narrowly spaced (measured periplasmic widths are $\approx 25$ nm[43]) so that the total membrane area is approximately twice the measured cell surface area, $A_{mem} \approx 2S_A$. Applying these simplifications and taking the ratio of Eq. (1) and Eq. (2) yields an expression for the cytoplasm-membrane density ratio $\kappa$:

$$\kappa \equiv \frac{\rho_{cyto}}{\sigma_{mem}} = \frac{\psi_{cyto} M_{prot}^{(tot)} + M_{RNA}^{(tot)}}{V} \times \frac{2S_A}{\psi_{mem} M_{prot}^{(tot)}} = \frac{2(\psi_{cyto} + \frac{M_{RNA}^{(tot)}}{M_{prot}^{(tot)}})}{\psi_{mem}} \times \frac{S_A}{V}. \tag{3}$$

This expression makes explicit the dependence of $\kappa$ on both the surface-to-volume ratio $S_A/V$, a quantity proposed as a "state variable" for bacterial morphogenesis[22], and the RNA-to-protein ratio $M_{RNA}^{(tot)}/M_{prot}^{(tot)}$, a key determinant of growth rate and the core variable of the ribosomal growth law[44]. The density ratio $\kappa$ thus acts as a scaling factor linking cellular geometry to proteome partitioning and macromolecular composition, embedding both geometric and biosynthetic constraints.

Given that proteins are partitioned among three compartments (Fig. 1a, $\psi_{cyto} + \psi_{mem} + \psi_{peri} \approx 1$) and that the RNA-to-protein ratio is directly related to the ribosomal fraction of the total proteome (termed the *ribosomal proteome allocation $\phi_{rib}$*) with a proportionality constant $\beta$ (see Supplementary Note 4), Eq. (3) can be rewritten as

$$\frac{S_A}{V} = \frac{\kappa \psi_{mem}}{2(1 - \psi_{mem} - \psi_{peri} + \beta \phi_{rib})}, \tag{4}$$

which expresses the surface-to-volume ratio in terms of the proteome partitioning and the proteomic allocation towards ribosomes.

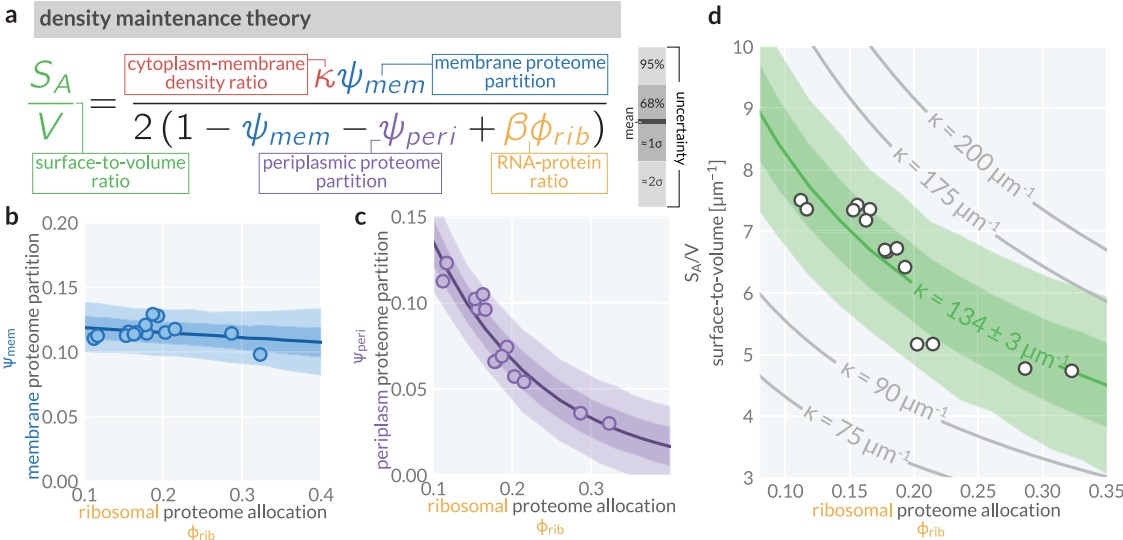

**Fig. 3 | A density maintenance theory quantitatively predicts scaling between surface-to-volume ratio ($S_A/V$) and proteome allocation towards ribosomes $\phi_{rib}$. a** The density maintenance theory as derived in the text with the density ratio $\kappa$, membrane proteome partition $\psi_{mem}$, periplasmic proteome partition $\psi_{peri}$, and RNA/protein ratio $\beta\phi_{rib}$ highlighted in red, blue, purple, and gold, respectively. **b** The empirical linear (approximately constant) relationship between membrane proteome partition and ribosomal proteome allocation. **c** The empirical exponential relationship between periplasmic proteome partition and ribosomal proteome allocation. **d** Predicted scaling of surface-to-volume $S_A/V$ on the ribosomal proteome allocation $\phi_{rib}$ for different values of the density ratio $\kappa$ (gray lines). Green curves correspond to the fit of Eq. (4) to our data, using the empirical relationships shown in b and c. Lines correspond to the mean value of the posterior predictive distribution. Shaded bands correspond to the 95% (pale, approximately $2\sigma$) and 68% (light, approximately $1\sigma$) credible regions of the posterior predictive distribution for each quantity. White-faced points correspond to our direct measurements of allocation parameters via mass spectrometry and $S_A/V$ via microscopy. All biological replicates are shown.

## Surface-to-volume and proteome align with model predictions

Equation (4), schematized in Fig. 3a, provides a quantitative prediction for how the cellular geometry, specifically the surface-to-volume ratio $S_A/V$, scales with the composition and localization of the proteome under the assumption that the cytoplasm-to-membrane density ratio $\kappa$ remains constant. In particular, this framework predicts that $S_A/V$ is determined by the proteome fraction partitioned to membranes ($\psi_{mem}$) and the periplasm ($\psi_{peri}$), and the ribosomal proteome allocation ($\phi_{rib}$), linking cell size control directly to proteome composition.

We sought to determine whether the observed $S_A/V$ follows the expected non-linear dependence on $\phi_{rib}$ as prescribed by Eq. (4). Theoretical and experimental dissections of the ribosomal growth law[3,7,8,11] have established that the ribosomal proteome fraction $\phi_{rib}$ increases from approximately 5–30% of the total proteome from slow to fast growth conditions, respectively (Supplementary Fig. 1a). If $\phi_{rib}$ serves as a control variable for growth, then our density maintenance model predicts a corresponding non-linear scaling of $S_A/V$. However, accurately making this prediction requires empirical knowledge of how $\psi_{peri}$ and $\psi_{mem}$ change as a function of $\phi_{rib}$. Using our mass spectrometry measurements, we characterized these dependencies and found that $\psi_{mem}$ remains nearly constant, while $\psi_{peri}$ exhibits an approximately exponential decline as a function of $\phi_{rib}$ (Fig. 3b, c). Using a Bayesian inferential model to quantify our uncertainty (Methods and Supplementary Note 3), we parameterized these relationships and integrated them into our theoretical predictions. We then evaluated Eq. (4) under different candidate values for $\kappa$ and found that the predicted $S_A/V$ exhibits a strong non-linear dependence on $\phi_{rib}$ (Fig. 3d, gray lines), covering a physiologically plausible range of values for $S_A/V$ between ≈3 and 10 $\mu m^{-1}$. Using a Bayesian statistical model, we inferred the best-fit value of $\kappa = 134 \pm 3\ \mu m^{-1}$ to our direct measurements of $S_A/V$ (Fig. 3d, markers and green bands). We find that this model describes our observations with notable quantitative accuracy.

## ppGpp perturbations predictably alter cell geometry

Our analysis thus far suggests that the surface-to-volume ratio $S_A/V$ can be accurately described by a simple model defined by a constant cytoplasm-membrane density ratio $\kappa$ and a variable ribosomal allocation $\phi_{rib}$. This is confirmed from cells grown in different carbon sources with a range of growth rates, and therefore a range of $\phi_{rib}$. However, in contrast to other cell size models, cell geometry in our model (Eq. (4)) does not explicitly depend on the cellular growth rate nor the specifics of the environment, but only on the cellular composition. This provides a unique opportunity to test our model by composition perturbations.

Here we specifically perturbed the global regulator guanosine tetraphosphate (ppGpp)[6,45] which breaks the typical correlation between ribosomal content $\phi_{rib}$ and growth rate, allowing us to test whether $S_A/V$ follows composition (as Eq. (4) predicts) or growth rate (as conventional models suggest). To systematically alter ppGpp levels, we employed a genetic system developed by Büke et al.[46] that enables tunable expression of RelA and MeshI, enzymes that synthesize and degrade ppGpp, respectively (Fig. 4a). By titrating the expression of these enzymes in two growth conditions—glucose and glucose supplemented with casamino acids (glucose+CAA)—we directly measured the resulting changes in proteomic composition, partitioning, growth rate, and surface-to-volume ratio. Under Eq. (4), we would expect that increasing ppGpp levels would decrease $\phi_{rib}$, leading to an increase in $S_A/V$, while decreasing ppGpp should have the opposite effect (Fig. 4a, sliders).

Inducing the expression of MeshI, which decreases ppGpp, or RelA, which increases ppGpp, altered ribosomal proteome allocation $\phi_{rib}$ (Fig. 4b), growth rate (Fig. 4c), and surface-to-volume ratio (Fig. 4d) in the qualitatively predicted directions. Specifically, induction of MeshI or RelA altered the ribosomal proteome allocation $\phi_{rib}$ as anticipated, with corresponding shifts in growth rate and $S_A/V$ in both growth conditions (Fig. 4b). These results confirm that the ppGpp perturbations directly modulate ribosomal allocation and allow us to

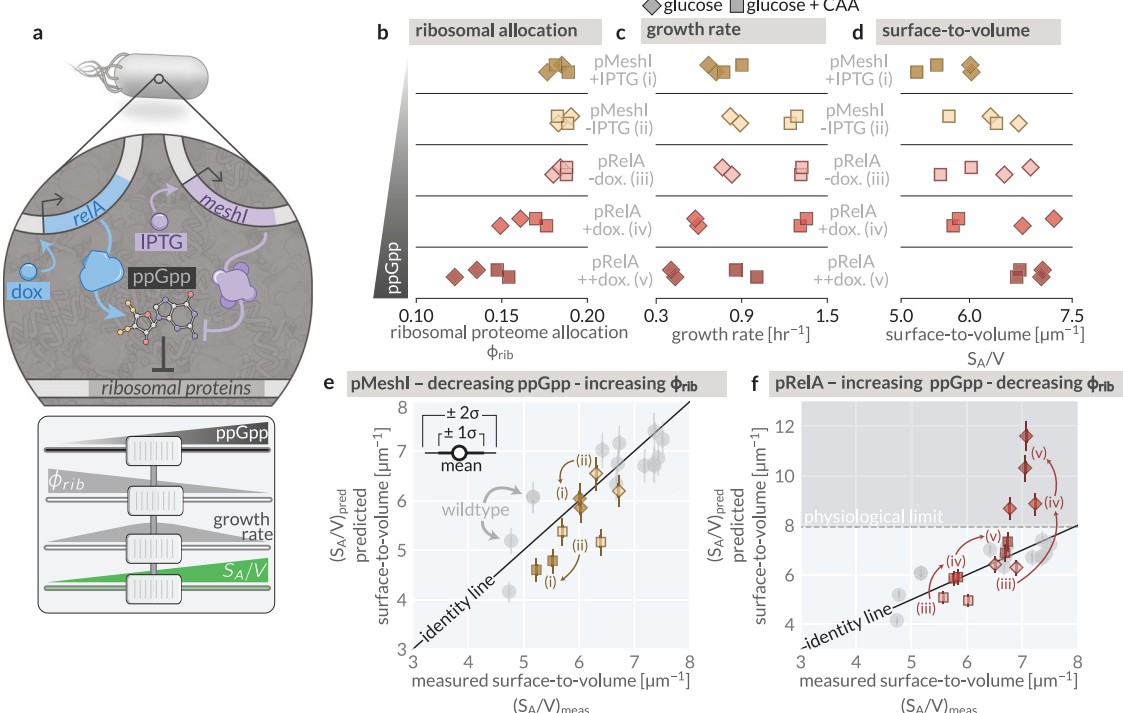

**Fig. 4 | Perturbing intracellular ppGpp predictably alters the surface-to-volume ratio. a** The genetic system as adapted from Büke et al.[46], allowing for inducible control over intracellular ppGpp concentrations (top). The expected effect of RelA or MeshI induction on ppGpp concentration, ribosomal proteome allocation, growth rate, and the surface-to-volume ratio (bottom). Observed effects of MeshI/RelA induction on **b** ribosomal allocation, **c** growth rate, and **d** surface-to-volume ratio in a glucose (diamonds) or casamino acid-supplemented glucose growth medium (squares). The predicted versus measured $S_A/V$ for MeshI (**e** gold markers) and RelA (**f** red markers) induction conditions. Gray circles correspond to the wildtype strain in different growth conditions. Thin and thick error bars correspond to the 95% (≈2σ) and 68% (≈1σ) credible regions of the posterior distribution for the surface-to-volume assuming a constant density ratio $\kappa$ inferred in Fig. 3c. MeshI was induced with IPTG ( − : 0 μM + : 100 μM,) and RelA was induced with doxycyline ( − : 0 ng/mL; + : 2 ng/mL, + + : 4 ng/mL).

assess their impact on the surface-to-volume ratio independent of the growth rate (Supplementary Fig. 7a).

To quantitatively compare the predicted and measured $S_A/V$ under ppGpp perturbations, we determined the partitioning and allocation parameters in Eq. (4) from mass spectrometry measurements (Supplementary Fig. 6) and assumed that $\kappa$ remains fixed at the inferred value of $\kappa = 134 \pm 3$ μm⁻¹ (Fig. 3d). For MeshI induction [Fig. 4e, (ii) → (i)], the predicted and measured $S_A/V$ are in strong quantitative agreement for both glucose and glucose+CAA conditions, with values falling within experimental variation of the wildtype strain. For RelA induction [Fig. 4f, (iii) → (iv) → (v)], we find that predicted and measured $S_A/V$ agree well within the glucose+CAA condition but diverge in the glucose-only condition at high RelA expression levels. In this scenario, Eq. (4) predicts $S_A/V \approx 12$ μm⁻¹, which deviates significantly from the measured values of $S_A/V \approx 7$ μm⁻¹ (Fig. 4d). We hypothesize that RelA induction in glucose pushes the system into a regime where maintenance of a constant density ratio is not physiologically possible. Indeed, assuming a constant proportionality between RNA mass and ribosomal content (Methods), we can empirically determine that the cytopasm-membrane density ratio under these conditions [Supplementary Fig. 7b] decreases to approximately 100 μm⁻¹. While not a direct measurement of the density ratio, this result is suggestive that the assumption of a constant density ratio breaks down due to a physical upper bound in $S_A/V \approx 8$ μm⁻¹ (dashed line). Given a rod-shaped bacterium, this $S_A/V$ corresponds to a cell width $w \approx 0.5$ μm, in line with the minimum width of *E. coli* that has been observed experimentally (Supplementary Fig. 2). Maintaining a constant density ratio for the high RelA induction conditions in glucose would require cells to adopt a width of ≈0.25 μm, well below this physiological limit. This establishes a boundary of validity for the density maintenance model, providing

insight into the range of conditions under which proteome composition alone can predict cellular geometry.

## Discussion

In this work, we analyze the interdependence between cellular composition, proteome localization, and cellular geometry. We propose a biophysical principle that lies at the center of this interdependence: macromolecular densities within the cytoplasm and the areal density of proteins in the cell membrane are maintained within a narrow range. Cell geometry changes as a byproduct of proteome partitioning to maintain these densities.

Motivated by the well-maintained ratio of densities, we derive a simple mathematical model that quantitatively relates the surface-to-volume ratio to the partitioning of the proteome between compartments and the functional composition of the proteome, specifically the proteome fraction composed of ribosomes. We conducted a suite of experiments measuring steady-state growth rate, cell size, and proteome composition across a broad range of conditions. To our knowledge, this is the only data set with simultaneous measurements of these quantities with sufficient statistical power to test our biophysical model. Using this dataset and genetic perturbations, we demonstrate that the model of density maintenance accurately predicts how the surface-to-volume ratio depends on steady-state proteome composition and localization. While further systematic and more direct measurements of density ratios under broad genetic perturbations remain an important direction for future studies, our findings demonstrate that cell composition and density control are major determinants of cellular geometry, more predictive than growth rate itself.

Beyond our own observations, we find that this picture stands in good agreement with the literature examining what does (and does

not) alter cell size across conditions. For example, Basan et al.[28] utilized the strong over-expression of a non-needed cytoplasmic protein to drastically change the composition. As anticipated by our theory, the authors report that the width and the average cell size increased considerably while total dry mass density was maintained. Furthermore, as our theory does not include any rate parameters or binding constants, we would expect its predictions to be independent of temperature. Indeed, this is consistent with previous studies showing that cell composition and size are both well-maintained across wide temperature ranges, while the growth rate is strongly temperature dependent[44,47-51]. Finally, while we focus in this work on *E. coli*, there is evidence that density maintenance may be a more general property across the microbial world. For example, recent work in *Corynebacterium glutamicum*[52], a Gram-positive bacterium, reveals a strong correlation between the surface-to-volume ratio and the RNA-to-protein ratio that is consistent with our theoretical predictions. Similarly, the methanogenic archaeon *Methanococcus maripaludis* demonstrates a fixed composition across growth conditions and, in line with our theory, a constant cell size[53]. In total, a hypothesis that cells prioritize the maintenance of macromolecular densities and do so through adjustments in cell size is supported by a litany of observations which have at times seemed incongruous.

Recently, Büke et al.[46] demonstrated that ppGpp directly altered average cell volume in a manner that was uncoupled from the bulk growth rate. While unequivocally establishing a relationship between ppGpp concentration and cell size, the precise mechanism underlying this connection remains unclear. Our hypothesis of density maintenance provides a natural explanation–intracellular ppGpp pools modulate ribosomal content by regulating the expression of ribosomal rRNA and protein genes, thereby altering cellular composition and, in turn, cell geometry. Other work by Harris & Theriot[22] has proposed that the surface-to-volume ratio is a quantity that cells actively monitor and homeostatically control through the coordination of volume and surface expansion. Our work builds upon this idea by providing a plausible biophysical principle by which $S_A/V$ can be regulated, as illustrated in Panel A of Fig. 5. Throughout the cell cycle, rod-shaped bacterial cells maintain a fixed width while their length increases, and so long as the length remains significantly larger than the width ($\ell \gg w$), the surface-to-volume ratio follows the simple relation $S_A/V \approx 4/w$. This implies that rather than directly monitoring both surface area and volume, cells could achieve robust control over $S_A/V$ simply by regulating their width. Under this framework, maintaining a constant density ratio $\kappa$ provides a natural feedback mechanism linking

biochemical composition to morphological regulation, ensuring that cell geometry is tuned to maintain homeostasis across different conditions.

This model raises two fundamental molecular questions: how could cells maintain density homeostasis, and how might this be coupled to width control? While the specific mechanisms underlying density maintenance remain unclear, the Rod complex represents a plausible candidate for such coupling. The Rod complex is a large protein assembly[54-56] found across the bacterial tree of life[57], which rotates about the long axis of the cell along the inner membrane, expanding the cell wall and, therefore, increasing the cell volume and surface area[58,59]. While lengthening the cell over the course of the cell cycle, the Rod complex also determines the width of the cell[55,60,61], thereby controlling the surface-to-volume ratio. As the Rod complex rotates through both the cytoplasmic and membrane environments, it could potentially be subjected to density-dependent forces that influence its activity. This raises the possibility that the Rod complex function might be modulated by local macromolecular densities, providing a mechanism for coupling cellular composition to geometric control. Consistent with this possibility, genetic perturbations of various Rod complex components have been shown to strongly affect cell size and shape homeostasis[61,62]. Irrespective of the molecular mechanisms at play, our predictive framework of density maintenance provides a quantitative foundation for understanding the relationships between size and composition and their physiological consequences.

Despite evidence that growth rate regulation and cell size control are uncoupled in various situations–such as through temperature variation– growth rate is commonly viewed as a control variable for bacterial physiology as a whole. However, we argue that growth should be thought of as an emergent property of the cellular physiology, as is cell size: Cell composition is set by the coordination of gene expression following from sensing of the cells' environment and its metabolic state. Growth rate emerges from the relative rates of metabolism and translation resulting from this composition[11]. Separately, as we have demonstrated in this work, the pressure to maintain macromolecular densities within the cytoplasm and membrane compartments strongly constrains cellular size. As a consequence, strong correlations between cell size and growth rate can emerge even without a direct causal link between them (Fig. 5b). Accordingly, approaches to understand cell physiology should not rely on growth rate as an explanatory process, but rather the fundamental physical and chemical limits that cells must obey and can plausibly biochemically measure.

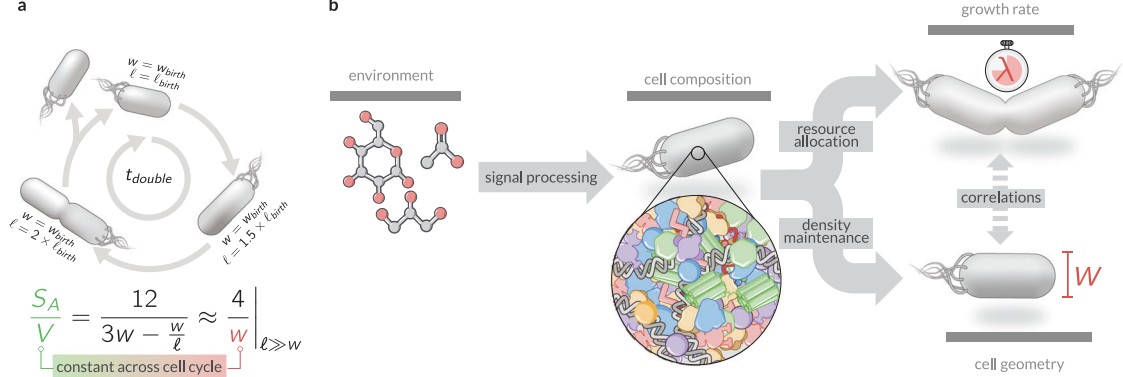

**Fig. 5 | Growth rate as an emergent property of compositional and geometric regulation. a** The cell length changes over the course of the cell cycle, approximately doubling the cell volume, but the cell width remains approximately constant within a given growth condition. The $S_A/V$ (bottom) of a rod-shaped cell is therefore approximately constant across the cell cycle. **b** Chemical details of the environment set the cellular composition through sensory pathways and integrated regulation of

gene expression. Given the cellular composition, the bulk growth rate is determined via the regulation of metabolic and translational fluxes, setting cellular composition. Simultaneously and following our density maintenance theory pressure to constrain macromolecular densities within the cytoplasm and membrane protein densities within the membrane determines cellular geometry.

## Methods

### Bacterial strains and cell husbandry

Experiments performed in this work were conducted using the *E. coli* K-12 strain NCM3722 supplied from the laboratory of Terence Hwa at UCSD, originally obtained from the laboratory of Sydney Kustu[63]. Perturbations of intracellular ppGpp concentrations were performed using a genetic system as described in Büke et al.[46]. These plasmids (without fluorescent tags) were ordered from AddGene (pRelA' AddGeneID:175595; pMeshI AddGeneID:175594) and transformed individually into our lab stock of NCM3722 on appropriate selection conditions. All used strains are listed in Supplementary Table 2. Culturing plasmids was performed under either Ampicillin (pMeshI; 100 µg/mL) or Kanamycin (pRelA; 50 µg/mL) selection.

To ensure sample analysis at steady-state, cells were processed through three different cultivation steps before samples were taken. To start, "seed culture" was grown in Miller LB rich medium (Fisher Scientific, Cat. No. BP1426) from a single colony on an LB agarose plate. This seed culture was grown in a 37 °C waterbath shaker under constant aeration (shaking at 240 rpm) for several hours until the culture was saturated. This seed culture was then diluted at least three hundred fold into fresh LB media or a minimal phosphate buffer medium (5.7 mM $K_2SO_4$, 77.6 mM $KH_2PO_4$, 34.6 mM $KH_2PO_4$, 400 µM $MgSO_4$, 43.1 mM NaCl, 10 mM $NH_4Cl$) supplemented with a carbon source (either 30 mM acetate, 10 mM sorbitol, 10 mM glycerol, or 10 mM glucose) or mix of carbon sources (glucose + acetate or glucose + 0.1% w/v casamino acids). This culture, the "pre-culture condition", was then allowed to grow under constant aeration until an optical density $OD_{600nm} \approx 0.3 - 0.4$ (Thermo Scientific Genesys 30, 1-cm path length cuvette) was reached. This culture was diluted tenfold into 15 mL of fresh medium with the same composition, pre-warmed to 37 °C. This culture, the "experimental culture", was then grown in identical conditions as the pre-culture. Growth curves were obtained by regular $OD_{600}$ measurements while the culture remained between an optical density range of $OD_{600nm} \approx 0.04 - 0.5$. For strains with ppGpp perturbations, the seed culture was grown in a glucose-supplemented minimal medium. Once the seed culture reached an optical density $OD_{600nm}$ between 0.3 and 0.4, the culture was diluted two-thousand fold into a fresh, prewarmed glucose minimal medium supplemented with the appropriate amount of inducer, either doxycycline (dox, Sigma, Cat. No. D5207) or Isopropyl $\beta$- d-1-thiogalactopyranoside (IPTG, Goldbio Cat. No. 12481C5) for RelA and MeshI induction, respectively.

During the growth of the experimental culture, samples were taken and processed for cell size measurement and mass spectrometry as described briefly below. All cultivation steps were performed in glass tubes with at least four volumes of head space for sufficient gas exchange and aerobic growth.

### Quantification of cell size

From a steady-state culture, 2 µL was transferred onto a 1% agarose pad supplemented with isotonic minimal medium buffer base. After drying for 2–3 min, this pad was mounted on a slide, covered with a coverslip, and imaged under 100X phase-contrast microscopy using a Zeiss AxioVert 200M microscope outfitted with an AmScope MU1003 CMOS camera. Images were transferred to a backup server and were later processed using in-house image processing Python code, as described in Supplementary Note 2.

### Proteome quantification via mass spectrometry

Cultures were grown to a final $OD_{600nm} \approx 0.4 - 0.5$ and were then harvested for proteomic analysis. An aliquot of 12 mL of the steady-state culture was removed from the culture vessel and transferred to a 14 mL centrifuge tube. Cells were pelleted at 3000 × *g* for 10 min at 4 °C. All but 1 mL of the supernatant was carefully decanted and transferred to a clean 14 mL centrifuge tube. The pellet was resuspended in the residual 1 mL of supernatant and transferred to a 1.5 mL Eppendorf tube and pelleted at 3000 × *g* for 1 min at 4 °C. The supernatant was removed and added to the decanted supernatant, and thoroughly mixed. The $OD_{600nm}$ of this pooled supernatant was recorded to correct for cell loss. The pellet was then frozen at −80 °C for later proteomic analysis.

When ready for proteomic processing, pellets were removed from the −80 °C and thawed on ice. The pellets were then resuspended in a denaturation/reduction buffer (0.07M Tris-Cl, 5% (v/v) mercaptoethanol, 0.6% (w/v) SDS, and 15% (v/v) glycerol) by boiling for 10 min at 95 °C with intermittent vortexing. Debris was pelleted by centrifugation for 10 min (10,000 × *g*). Cleared supernatants were alkylated with 5 mM iodoacetamide and then precipitated with three volumes of a solution containing 50% acetone and 50% ethanol. Proteins were re-solubilized in 2 M urea, 50 mM Tris-HCl, pH 8.0, and 150 mM NaCl, and then digested with TPCK-treated trypsin (50:1) overnight at 37 °C. Trifluoroacetic acid and formic acid were added to the digested peptides for a final concentration of 0.2%. Peptides were desalted with a Sep-Pak 50 mg C18 column (Waters). The C18 column was conditioned with 500 µL of 80% acetonitrile and 0.1% acetic acid and then washed with 1000 µL of 0.1% trifluoroacetic acid. After samples were loaded, the column was washed with 2000 µL of 0.1% acetic acid, followed by elution with 400 µL of 80% acetonitrile and 0.1% acetic acid. The elution was dried in a Concentrator at 45 °C. Desalted peptides were resuspended in 0.1% Formic acid.

For each sample, 25 µg of desalted peptide samples were resuspended in 20 µL of 100 mM Triethylammonium bicarbonate solution and labeled with 16-plex TMTpro at a ratio 4:1 (TMT:peptide). Total reaction volume was less than 25 µL. The labeling reaction was quenched with a final concentration of 0.5% hydroxylamine for 15 min. Labeled peptides were pooled and acidified to pH ~2 using drops of 10% TFA. Excess TMT label was removed by re-running the pooled sample through a Sep-Pak 50-mg C18 column (as described above).

TMT-labeled peptides were resuspended in 0.1% formic acid analyzed on a Fusion Lumos mass spectrometer (Thermo Fisher Scientific, San Jose, CA) equipped with a Thermo EASY-nLC 1200 LC system (Thermo Fisher Scientific, San Jose, CA). Peptides were separated by capillary reverse phase chromatography on a 25 cm column (75 µm inner diameter, packed with 1.6 µm C18 resin, AUR2-25075C18A, Ionopticks, Victoria, Australia). Peptides were introduced into the Fusion Lumos mass spectrometer using a 180-min stepped linear gradient at a flow rate of 300 nL/min. The steps of the gradient are as follows: 6−33% buffer B (0.1% (v:v) formic acid in 80% acetonitrile) for 145 min, 33−45% buffer B for 15 min, 40−95% buffer B for 5 min, and maintain at 90% buffer B for 5 min. The column temperature was maintained at 50 °C throughout the procedure. Xcalibur software (v.4.4.16.14) was used for the data acquisition, and the instrument was operated in data-dependent mode. Advanced peak detection was disabled. Survey scans were acquired in the Orbitrap mass analyzer (centroid mode) over the range 380−1400 m/z with a mass resolution of 120,000 (at m/z 200). For MS1, the normalized AGC target (%) was set at 250, and the maximum injection time was set to 100 ms. Selected ions were fragmented by collision-induced dissociation (CID) with normalized collision energies of 34, and the tandem mass spectra were acquired in the ion trap mass analyzer with the scan rate set to "Rapid". The isolation window was set to the 0.7 m/z window. For MS2, the normalized AGC target (%) was set to "Standard" and the maximum injection time to 35 ms. Repeated sequencing of peptides was kept to a minimum by dynamic exclusion of the sequenced peptides for 30 s. The maximum duty cycle length was set to 3 s. Relative changes in peptide concentration were determined at the MS3 level by isolating and fragmenting the five most dominant MS2 ion peaks.

All raw files were searched using the Andromeda engine embedded in MaxQuant (v2). Reporter ion MS3 search was conducted using TMTpro (16-plex) isobaric labels. Variable modifications included

oxidation (M) and protein N-terminal acetylation. Carbamidomethyl (C) was a fixed modification. The number of modifications per peptide was capped at five. Digestion was set to tryptic (proline-blocked). Database search was conducted using the UniProt proteome—`Eco-li_UP000000625_83333`. The minimum peptide length was 7 amino acids. 1% FDR was determined using a reverse decoy proteome.

To calculate the proteome mass fraction of each mapped protein, we utilized the peptide feature information in MaxQuant's `evidence.txt` output file. Each row of the `evidence.txt` file represents an independent peptide and its corresponding MS3 reporter ion measurements. Peptides without a signal in any of the TMT channels were excluded. Peptide measurements were assigned to a protein based on MaxQuant's "Leading razor protein" designation. For each individual peptide measurement (i.e., each row in the evidence table), the fraction of ion intensity in each TMT channel was calculated by dividing the "Reporter ion intensity" column by the sum of all reporter ion intensities. To correct for loading differences between the TMT channels, each reporter ion channel was then normalized by dividing the fraction of ion intensity in each channel by the median fraction for all measured peptides (i.e., the median value for each column). This normalization scheme ensures that each individual peptide measurement is equally weighted when correcting for loading error. To calculate proteome mass fractions, the MS1 precursor ion intensity of each peptide measured (the "Intensity" column in the `evidence.txt` table) was distributed between the individual MS3 reportion channels according to the loading-normalized value described. Protein-level ion intensities were then calculated for each TMT channel by summing together all peptides ion intensities for each protein.

### Quantification of total RNA and protein masses
The total RNA and total protein measurements needed to calculate the compartment densities in Fig. 2 were measured from cultures independent of those used for mass spectrometry and size quantification.

Briefly, cells were cultured in a basic buffer medium as described above and harvested at an $OD_{600nm} \approx 0.5$. The Biuret method[64] was used for total protein quantification. For each culture, an aliquot of 1.8 mL of cell culture was transferred to a clean 2 mL test tube and was pelleted at 16,000 × $g$ for 1 min. The supernatant was removed, and the $OD_{600nm}$ was measured to correct for cell los. The pellet was washed with 1 mL of ddH$_2$O and pelleted again at 16,000 × $g$. The supernatant was again carefully removed, and the $OD_{600nm}$ was determined. The pellet was resuspended in 200 μL of ddH$_2$O and transferred to a −80 °C freezer for 10 min. Once frozen, pellets were transferred to an ice bath, and 100 μL of 3M NaOH was added to the pellets, followed by boiling at 100 °C for 5 min, followed by a 5 min cool-down at room temperature. A 100 μL aliquot of 1.6% CuSO$_4$ was added to the disrupted cell pellet, shaken vigorously, and incubated at room temperature for 5 min. The slurry was then centrifuged at 16,000 × $g$ for 3 min and the $OD_{555nm}$. This was converted to a concentration using a calibration curve of known BSA concentrations.

Total RNA quantification was performed as described previously[65]. Briefly, a 1.5 mL aliquot from the steady-state culture was removed and transferred to a 2 mL test tube. Cells were pelleted at 16,000 × $g$ for 3 min. The supernatant was carefully removed, and the $OD_{600nm}$ was determined to correct for cell loss. The pellet was washed twice with 600 μL of 0.7 mM HClO$_4$ at 4 °C, with the supernatants pooled and their combined $OD_{600nm}$ determined. The washed pellet was resuspended in 300 μL of 0.3 M KOH and was incubated at 37 °C with shaking for 1 h. RNA was extracted from the digested cell solution by adding 100 μL of 3M HClO$_4$, and debris was pelleted at 16,000 × $g$ for 1 min. The supernatant was removed and transferred to a new 2 mL test tube. The debris pellet was washed twice with 550 μL 0.5 M HClO$_4$ kept at 4 °C, and supernatants were pooled. The UV absorption of the supernatant at 260 nm was measured, and total RNA was calculated using a conversion factor of 31 μg/A$_{260nm}$.

Total cell counts for each growth condition were quantified using flow cytometry. Briefly, cells were grown in the appropriate growth medium to an $OD_{600nm} \approx 0.2 - 0.4$, and 500 μL was transferred to a sterile 1.5 mL Eppendorf tube. A volume of fixation buffer (PBS with 0.9% NaCl and 0.12% formaldehyde) was added to the cells and gently mixed. To 810 μL of buffered DAPI (0.9% NaCl with 1 μg/mL DAPI), 100 μL of the above cell sample was added. This was incubated on ice for 3 min. Samples were then injected and analyzed in a BD FAC-Symphony A5 Cell Analyzer with a SSC gain of 1000 and a flow rate of 1 μL/s. Signal was collected for 100 s per sample. All objects which were DAPI positive were noted as individual cells. We found that applying an automated gating procedure (described in Razo-Mejia et al.[66]) did not significantly influence our estimate of cells per unit biomass.

### Quantification of cellular growth rate
During the steady-state growth, the $OD_{600nm}$ was measured at regular intervals using a Thermo Scientific GENESYS 30 Spectrophotometer. For each growth cycle, the growth rate was determined by performing a linear regression on the log-transformed optical density measurements within the linear range of the spectrophotometer (0.04–0.5 for our specific spectrophotometer). This was performed on a per-replicate basis, and the results of the fitting were saved and collated as a CSV file. The linear regression was performed using the `linregress` function in the Python `scipy.stats` library (version `1.10.0`) using default parameters.

### Localization annotation of proteomic data
The localization of each protein detected within our mass spectrometry measurements was determined using the localization criteria defined in Babu et al. 2018[31]. Proteins detected in our experiments, yet not annotated or located within the Babu et al. annotation form, were dropped from all subsequent analysis. In total, this occurred only for 14 proteins, accounting for at most 0.1% of the total proteome mass.

### Calculation of compartment densities
In this work, we used bulk measurements of total protein and total RNA coupled with partitioning information from our quantitative mass spectrometry measurements to estimate the protein masses and densities within each compartment for our wild-type *E. coli* strains (Supplementary Fig. 5a). For each compartment, the total protein mass in each compartment $M_{prot}^{(compartment)}$ was calculated as

$$M_{prot}^{(compartment)} = \psi_{compartment} M_{prot}^{(tot)} \qquad (5)$$

where $\psi_{compartment}$ denotes the proteome partitioning to that compartment and $M_{prot}^{(tot)}$ denotes the total protein mass per cell, determined from bulk measurements. However, we did not directly measure the total protein per cell for each mass spectrometry measurement. To do so, we instead assumed that the total protein per cell scaled exponentially as a function of the growth rate $\lambda$, and used the resulting fit to estimate the total protein per cell for samples subjected to mass spectrometry quantification, given their measured growth rate. To perform this fit, we used a Bayesian inferential model to compute the posterior distribution over the parameters of the exponential fit as well as the expected homoskedastic measurement error. Details of this inference, along with the choice of priors and appropriate transformations, are given in the Supplementary Note 3. The result of this fit is shown in Supplementary Fig. 5b.

For the cytoplasmic density, we made the well-justified assumption that the majority of cytoplasmic mass is composed of protein and RNA[4,42] with DNA contributing a negligible fraction of the mass. Thus, the cytoplasmic density was defined by the proteome mass of the cytoplasm and the total cellular RNA. As in the case of total protein, we did not directly measure total RNA for every sample subjected to mass spectrometry. We assumed that the total RNA mass per cell $M_{RNA}^{(tot)}$ also

scaled exponentially with the cellular growth rate and used the resulting fit to estimate the RNA for our samples subjected to mass spectrometry. We also used a Bayesian inferential model for this estimation, described in the Supplementary Note 3. The resulting fit is shown in Supplementary Fig. 5c.

### Bayesian inference

Throughout this work, we use Bayesian inferential models to quantify our uncertainty in our estimate of a quantity, such as the compartment densities or the density ratio parameter $\kappa$. While details change from model to model, the same basic approach applies, which we outline here. We adopt a Bayesian definition of probability and seek to evaluate the posterior probability distribution $g(\theta\,|\,y)$ of a parameter $\theta$ conditioned on a set of measurements $y$. Using Bayes' theorem, this can be computed as

$$g(\theta\,|\,y) = \frac{f(y\,|\,\theta)g(\theta)}{f(y)}, \qquad (6)$$

where $g$ and $f$ denote probability density functions over parameters and data, respectively. For the data observed in this work, we used a Gaussian distribution for the likelihood function $f(y\,|\,\theta)$ for the parameter(s) of interest with a model-dependent mean $\mu$ and standard deviation $\sigma$. The choice of prior distribution $g(\theta)$ was dependent on the precise parameter being inferred (see Supplementary Note 3). For this work, the denominator $f(y)$ of Eq. (6) was treated as a normalization constant and was therefore neglected in the estimation. All statistical modeling and parameter inference was performed using Markov chain Monte Carlo (MCMC). Specifically, we used Hamiltonian Monte Carlo sampling as is implemented in the Stan programming language[67]. To assess the accuracy of our inference, we evaluated the posterior predictive distribution by randomly sampling values from the likelihood function $f(y\,|\,\theta)$ for each MCMC sample of $\theta$. From this distribution, we then calculated the arithmetic mean and the bounds of the 68% and 95% percentiles, which correspond to approximately $1\sigma$ and $2\sigma$ of a Gaussian distribution. All statistical models are defined as `Stan` files are available on the paper's GitHub repository github.com/cremerlab/density_maintenance accessible via DOI: 10.5281/zenodo.10048570.

### Calculation of density ratios for ppGpp perturbations

In Supplementary Fig. 7b, we report the estimated cytoplasm-membrane density ratio for each ppGpp perturbation measurement. To do so for the wildtype strains, we used our inferred trends in the per-cell total protein $M_{prot}^{(tot)}$ and total RNA $M_{RNA}^{(tot)}$ masses as a function of growth rate. However, for our MeshI and RelA perturbation studies, these quantities do not necessarily scale in the same way with the growth rate. Thus, to calculate the cytoplasm-membrane density ratio, we made the well-motivated approximation that the total RNA mass can be calculated as

$$M_{RNA}^{(tot)} \approx \overbrace{\beta\phi_{rib}}^{\sim\frac{M_{RNA}^{(tot)}}{M_{prot}^{(tot)}}}M_{prot}^{(tot)}, \qquad (7)$$

where $\beta$ is a constant of proportionality that can be directly calculated[3] (Supplementary Notes). Using this approximation, the cytoplasm-membrane density ratio can be directly computed from the measured partitioning and allocation parameters determined via mass spectrometry and cell size measurements via

$$\frac{\rho_{cyto}}{\sigma_{mem}} \approx \frac{\psi_{cyto}M_{prot}^{(tot)} + \beta\phi_{rib}M_{prot}^{(tot)}}{\psi_{mem}M_{prot}^{(tot)}} \times \frac{A_{mem}}{V_{cyto}} = \frac{\psi_{cyto} + \beta\phi_{rib}}{\psi_{mem}} \times \frac{2S_A}{V - w_{peri}S_A}, \qquad (8)$$

where $A_{mem}$ is the total membrane area, $V_{cyto}$ is the cytoplasmic cell volume, $S_A$ is the cellular surface area, and $w_{peri}$ is the average width of the periplasmic space.

### Reporting summary

Further information on research design is available in the Nature Portfolio Reporting Summary linked to this article.

## Data availability

Raw microscopy images generated in this study have been deposited in the Stanford Data Repository accessible via DOI: 10.25740/ws785mz0287. The mass spectrometry proteomics data have been deposited to the ProteomeXchange Consortium via the PRIDE partner repository with the dataset identifier PXD071222. All other experimental data are available on the paper's GitHub repository DOI: 10.5281/zenodo.10048570 accessible via github.com/cremerlab/density_maintenance. In this work, we also integrated a large collection of datasets from the literature to evaluate the consistency of our measurements and guide our analysis. The datasets used from these studies are provided in the GitHub repository, and the corresponding references are as follows: proteomic measurements[33,68–74]; cell size measurements[18,28,75–78]; total protein, total RNA, and RNA/protein[3–5,14,18,28,29,35,42,79–85].

## Code availability

All Python code, Stan probabilistic models, and processed data sets are available on the paper's GitHub repository DOI:10.5281/zenodo.10048570 accessible via github.com/cremerlab/density_maintenance.

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

## Acknowledgements

We thank Markus Arnoldini, Rohan Balakrishnan, Nathan Belliveau, Avi Flamholz, Akshit Goyal, Mathis Leblanc, Shaili Mathur, Manuel Razo-Mejia, Tom Röschinger, Gabe Salmon, Masaru Shimasawa, Jan Skotheim, Alfred Spormann, and the three reviewers for extensive discussion and critical feedback on the manuscript. We also thank Ferhat Büke and Sander Tans for providing access to data from their recent work[46]. G.C. acknowledges support by the NSF Postdoctoral Research Fellowships in Biology Program (grant no. 2010807). M.C.L. acknowledges support from the Chan Zuckerberg BioHub through a Collaborative Postdoctoral Fellowship. This research was funded in part by the National Institute of General Medical Sciences, National Institutes of Health, grant number 1R01GM149611. J.C. also acknowledges support via a Terman Fellowship from Stanford University, USA.

## Author contributions

G.C., R.de.S., and J.C. conceptualized and designed the study. G.C., R.de.S., R.S., and M.C.L. performed experiments for the original submission of the study. G.C. performed experiments for the published version of the study. M.C.L. acquired mass spectrometry measurements, and M.C.L. and G.C. analyzed mass spectrometry data. G.C. and J.C. formulated the mathematical model. G.C. performed data analysis, developed the software, and formulated and executed statistical models. G.C. created illustrations and data visualizations in the figures. G.C. and J.C. wrote the paper, supported by all authors.

## Competing interests

The authors declare no competing interests.
