## [Transparent Peer Review file · Nature Communications]

Maintenance of Cytoplasmic and Membrane Densities Shapes Cellular Geometry in *Escherichia coli*

Corresponding Author: Professor Jonas Cremer

Version 0:

Reviewer comments:

Reviewer #1

(Remarks to the Author)

Overall assessment

This work aims to explain how the RNA-to-protein ratio controls the surface-to-volume ratio of *E. coli* as a consequence of constant protein densities in the cytoplasm and on the membranes. It is an important open question to understand how cellular-level features, such as cell geometry and growth rate, emerge from the microscopic properties, such as the macromolecular composition. The authors revisited more than 30 published datasets, collected new data, and found that they generally support the main idea of this work. I agree with the authors that the protein densities could serve as a fundamental physical constraint to give rise to some general relationships, as discussed in this work. I appreciate the authors' efforts to pool the published datasets. However, I do have a few major concerns listed below, and I think the authors should address them for further consideration of the publication.

Major issues

1. The authors should discuss in more detail the generality/universality of their theory, formulated as equation 5. This relation seems to hold better across different nutrient conditions, as the authors tried to demonstrate. Specifically, the authors tried to show that RNA/protein ratio is the main control variable, if not the sole one, of the surface-to-volume ratio. However, my question is, is this theory also applicable to a broader range of conditions that *E. coli* experience in both lab and natural conditions? For example, when the surface-to-volume ratio is significantly perturbed by antimicrobial compounds (Nonejuie et al., PNAS 2013) or genetics tools (Zheng et al., PNAS 2016), does the change in RNA/protein ratio still follow equation 5? Similarly, for equation 7, which links the cell width to RNA/protein, some of the perturbations mentioned above could largely affect the cell width, so do they also alter the RNA/protein accordingly? If not, the authors need to clearly point out that their theory applies only to a limited range of conditions.
2. I am not very convinced by some of the fitting results. For example, the linear fitting in Fig. 2B does not seem very good, as the data scatters from ~0.05 to ~0.18, almost varying by three folds. The authors need to show the goodness of fit and discuss the likelihood that ϕ_{mem} stays constant across the conditions. Similar issues apply to other fittings, such as Figs. 1C&1D, Figs. 2C&2D, and Figs. 4A-4C.
3. In the ppGpp test, to make the argument that S/V follows a similar relation against RNA/protein as shown in Fig. 3D, the authors need to show that ϕ_{mem} and ϕ_{peri} also correlate with RNA/protein in a similar manner to that in Figs. 2B & 2C.
4. While the authors included a lot of published datasets, it is hard to distinguish which datasets are used in different plots. For example, the legends of the figures in Figs. 1A and 1B are mixed. It will be helpful for readers to assess the results if the legends can be separated for different plots. Another question is related to Fig. 2D, which uses much fewer published datasets in the plot. In principle, because the datasets in Figs. 1A and 1B all measured the growth rates, S/V can be recalculated from the volume values in Fig. 1B, and plotted against RNA/protein values that fall within a similar growth rate range. This could double-check the robustness of equation 5, although different datasets might use different strain backgrounds, media, and other lab conditions.

Minor issues

1. Can the authors justify the use of Bayesian inferential model and the need of probability distributions of RNA/protein? It seems that simple means such as standard errors is sufficient to evaluate the uncertainties of the measurements.
2. Typo in line 51: "Wiving."
3. Typo in line 15: "Escherichia. coli" should be "Escherichia coli."

(Remarks on code availability)

The Python codes provide detailed README files and protocols, which should be informative enough for interested readers to re-run the codes and test the results.

Reviewer #2

(Remarks to the Author)

The authors have compiled an impressive set of data from publications spanning decades. As far as I am aware, the authors are the first to integrate proteomics data sets together with cell size measurements in order to demonstrate that membrane protein density and cytoplasmic dry mass density are tightly constrained in *E. coli*. This is an interesting and important finding on its own.

Next, the authors claim that given these constraints, variations in ribosomal content must give rise to variations in cell size. The authors claim that this explains correlations between growth rate, cell dimensions, and ribosomal content. This is an interesting and controversial claim, and an intriguing alternative to existing hypotheses. Whether this is a consequence of a system that coordinates composition and cell size (as the authors propose), or is instead simply a correlation arising from the cell regulating other unrelated processes (a spandrel, defined in Wikipedia as "a phenotypic characteristic that is a byproduct of the evolution of some other characteristic, rather than a direct product of adaptive selection" – also see ref 24 in the MS) has not been clearly established by the authors.

The authors have convincingly demonstrated correlations; however, concluding that correlations must reflect insightful phenomenological growth laws is not straightforward because it is important to distinguish two possibilities: a) there is causality behind the correlation, or b) the observed correlation is simply a spandrel (a null interpretation). An example of a phenomenological finding that has provided real biological insight is the work by Scott et al (Science 2010, reference 9 in the MS). Scott et al. use a correlation between RNA/protein ratio and growth rate to establish the theory of proteome allocation. Importantly, Scott et al. challenged their own interpretation in various ways: by artificially inflating ribosome numbers with chloramphenicol, and by overexpressing a useless protein to alter proteome allocation. These tests were essential in helping to establish that resource allocation is indeed an essential principle underlying the correlation between RNA/protein ratio and growth rate. For the present study, such tests are lacking. Is there an experiment that can disprove the proposed hypothesis? Such a test that can help distinguish whether the authors are demonstrating a meaningful regulatory link between RNA/protein ratio and cell size.

For instance, one possible "null interpretation" would be that dry mass per cell volume is maintained by a system that monitors osmolarity (e.g., membrane channels). Furthermore, this system could easily be imagined as highly robust and completely indifferent to cell geometry, growth rate, and the RNA/protein ratio. This strikes me as a much simpler explanation. In the absence of an independent test of the hypothesis that demonstrates a direct link, it remains possible that the correlation observed is not a growth law but actually a consequence of a very robust macromolecular density homeostasis system that coincidentally occurs together with a system that couples cell size with (for instance) DNA replication timing. In that case, the authors' proposal of a system that specifically couples cell size to density would be incorrect.

On the other hand, if cell density homeostasis indeed constrains cell geometry via an indirect mechanism, I would argue that the authors have actually discovered that cell geometry is an emergent property of cell density being rigorously controlled by osmolarity maintenance systems acting together with the systems that maintain cell shape (e.g. the Rod complex). Discovering that cell size is an indirect consequence of density regulation would be very interesting and an important contribution!

So in summary, I have two significant conceptual criticisms:

- 1) The authors have a hypothesis inspired by a correlation, but do not propose or perform an experiment that has a chance of disproving this hypothesis. There is a danger that the hypothesis is actually untestable.
- 2) The authors do not address the possibility that drymass density might be regulated independently of cell geometry. If the authors can demonstrate that cell size is a spandrel resulting from robust density maintenance, that would nicely address this criticism!

Other suggestions:

The authors rightly point out that cell size and growth rate can be readily decoupled (as seen in Buke et al (ref 70) and Basan et al (ref 21)), as can growth rate and ribosomal abundance (as seen in Scott et al., and many other publications). The authors propose instead that cell composition is a "major determining factor of cell size control." The authors provide some support for this notion by modulating ppGpp and observing that the RNA/protein ratio and S/V ratio change in a manner predicted by their theory (Figure 3). However, it looks like the authors are using only 5 data points. As mentioned above, many similar experiments have been performed in which RNA/protein ratio is decoupled from growth rate and cell size: chloramphenicol titrations in Scott et al., and overexpression of "useless" proteins, which enlarges cells while slowing growth (Scott et al. and Basan et al.) Do results from such decoupling experiments also agree with the density maintenance theory? If so, the authors should show this explicitly in figures: e.g. one figure of cell size vs. growth rate that indicates decoupling after expressing useless proteins or after chloramphenicol treatment and then showing that even in these cases, the relationship between cell composition and cell size remains solid. If those literature data from decoupling conditions are already present in Figure 1, the authors should unpack such data as it would better support the proposal that cell

composition is a “major determining factor of cell size control” even when growth rate is decoupled from cell size and cellular composition. This would be a much more convincing illustration than the 5 data points from the RelA/Mesh1 experiments in Figure 3.

Page 2: “Wiving matter” (should be “Living matter”)

Page 2: “As there are enumerable interfacial interactions” – do the authors mean “innumerable interfacial interactions”?

(Remarks on code availability)

Reviewer #3

(Remarks to the Author)

The authors propose an elegant and small model on what determines the surface area-to-volume ratio of E. coli cells. They contrast their hypothesis, that the ratio of membrane density and cytosolic density are kept constant, with an alternative explanation that assumes the cell geometry is set as a function of growth rate. This is an interesting viewpoint and some of the data that the authors provide is convincing. However, I think that major revisions are necessary to convey this message clearly and avoid all appearances of circular reasoning. I have listed the points that I would suggest to revise below.

Major points

I'm almost sure that some of the posterior probability intervals that are being reported are not the relevant ones. For example, the authors note in line 69 that “both densities are remarkably tightly constrained with median values of $p_{\text{cyt}} = 287.09 +5.26 -5.21 \text{ fg / fL}$ and $\sigma_{\text{mem}} = 2.7 +0.4 -0.3 \text{ fg / } \mu\text{m}^2$ [Fig. 1(E)] where the sub- and super-scripts denote the lower and upper bounds of the 95% credible regions”

This could indeed be, exactly as the authors write, the 95% posterior prob. interval on the median density. However, when the authors want to claim that the densities are tightly constrained it is not enough to show that the posterior on the median is tightly constrained, as this is mostly a reflection of the number of datapoints. Instead, it would be better to simply report the variance on the measured densities, which seems to be much larger than the reported error-bars.

This is even clearer in line 106, where the authors claim “First, we observe that the membrane proteome fraction is a fixed quantity at $\phi_{\text{mem}} = 0.131 +0.006 -0.006$ (blue lines), suggesting that while the expression of individual membrane components may vary across conditions, the total membrane protein fraction is fixed.”

Given the shown datapoints, it seems quite a ridiculous claim that the fraction is a “fixed quantity” within a range of 0.012, while the figure clearly shows datapoints ranging from 0.05 to 0.15, and while most datapoints fall out of the reported 95% posterior prob. interval. Again, it could indeed be that the reported range is a reasonable uncertainty region on the estimated average fraction, but this does not say that the membrane proteome fraction is fixed.

I suggest that a relevant measure of the variation is reported, and that the claims are re-worded to clearly state that there is still significant variation, even though the quantities seem somewhat stable across conditions and datasets.

Relating to this, maybe the authors can provide plots in which the data of Figures 2B, 2C is stratified per dataset as a supplemental figure. To me, it seems that the two newest (and maybe most reliable datasets), i.e. from the authors themselves and from Mori et al. (2021), show a negative trend of the membrane proteome fraction with the ribosomal proteome fraction. In the authors' dataset, only the last point breaks this negative trend. However, it is not easy to judge this from this figure because the datasets are all overlaid, and the plot is therefore quite crowded. All in all, I'm not too convinced by the authors' claims that ϕ_{mem} is a fixed quantity, but I also think this is not strictly necessary for the authors' main conclusion to hold.

I suggest to remove the data-points of older studies in Figure 2D, and keep only the datapoints that have been gathered by the authors themselves. It would of course be good to re-use data if there was sufficient information present, however, if I understand it correctly, these datasets only report surface-to-volume ratios and growth rates. It is quite a stretch to test the authors' theory with that data. Because it requires the reader to accept that:

1 ϕ_{mem} is well-described by the constant blue line in Figure 2B. The blue line doesn't even describe the data very well on which it was fitted, let alone other data. So this is quite a stretch.

2 ϕ_{peri} is well-described by the purple line in Figure 2C. Even though this fit seems a bit more convincing, remember that the data used in Figure 2D is not shown in this plot because there is no proteome information for these older datasets.

3 The empirical relation between growth rate and RNA/prot-ratio describes the data in these older datasets very well.

Although this relation is the most established, it is clear from Figure 1A that there is plenty of variation left around the fitted line.

All in all, I find it impossible to assume all this and see this as independent proof of the authors' theory.

Since the datapoints from these older datasets are based on SA/V and growth rate only, the plotted points are in fact just a replotting of the known relation between SA/V and growth rate. Although it is in principle interesting that the authors can reproduce this relation using their theory, it should not be presented as proof of their more complicated theory.

I would suggest that the authors also add a scatterplot to Figure 2 of measured SA/V versus predicted SA/V according to their theory for their data, where they use the data from 2B and 2C directly (i.e. without using the fitted relations). Based on the figures, it seems that these predicted and measured values would match fairly nicely. In fact, I think this would make the theory a bit more general because it would show that Equation 2A holds independent of whether the membrane-proteome fraction is constant or not. Since the authors claim Figure 2 mostly “strongly support[s] a hypothesis that density maintenance

defines the cellular surface-to-volume.” this also should not depend on the behavior of the membrane proteome fraction.

The ppGpp-perturbation experiments are a very important contribution of this paper because they directly test the proposed theory. I do think Figure 3 can be made quite a bit stronger and more informative.

- First, why is the theoretical prediction (i.e. the green line) from Figure 2D not shown in Figure 3D?
- Second, although it is informative to see the posterior probability distributions in 3B and 3C I would advise also showing the actual datapoints. Currently, it is unclear how many datapoints these posteriors are based on, and it can also not be checked quickly by the reader if the posterior distributions look reasonable. I see that there are some data-lines shown in the Supplement, but since these are marked “our experimental measurements” I can’t tell which relate to Figure 2 and which to Figure 3.
- Third, and most importantly, if the authors want to show that the growth rate does not set the SA/V-ratio in these experiments, the reader needs to be provided with this information. However, no information on the growth rate is present, and we can thus not decide whether the relation based on a dependence of SA/V on growth rate is wrong, even though the authors claim to disprove this theory when they say: “In summary, these findings strongly supports the claim that cell geometry is set by the cell composition, and not the details of the particular growth condition.” So this information on the growth rate needs to be added, ideally in a plot that shows that the new datapoints don’t fall on the empirical growth rate vs SA/V-line.
- The points brought up in the discussion by the authors on temperature affecting growth rate without affecting cell geometry or composition, as well as the results from Basan et al. showing that changing RNA/prot by overexpressing a useless cytosolic protein affects cell size while the density is maintained should be brought up earlier. Showing these results and comparing them to their prediction if data is available or measuring it themselves would be a great support to their claim.

In Figure 4 I am doubting if the RNA/prot ratios that are shown for the old datasets are measured or again inferred as in Figure 2 using the ribosome growth law. In the case of the latter, this should be noted in the text.

This relates to a more general point, which is that it would be better if the authors always state more clearly what information is directly measured, inferred from proteomics, or inferred from empirical relations. I am convinced that the authors’ findings are non-trivial and interesting, but reading the current version of the manuscript, one is easily led to believe that everything is circular and trivial.

Generally, the authors claim in the Abstract that their work “provides a candidate biochemical mechanism for how cell size control is manifested.” I think this promise is not delivered upon, and I would suggest removing this from the Abstract. The theory that the authors propose is quite far from being a biochemical mechanism, as it is not clear how this density is maintained. For example, is the cell actively controlling the densities, or are the densities automatically kept constant because the corresponding area and volumes just expand because new proteins are made and take up space? Relating to that, I don’t think the authors can claim that they have shown “Stringent control ...” in the title. Whether the constant densities are due to stringent control or due to emergent biophysical properties is an interesting question that cannot be claimed solved, but rather should be addressed in the Discussion.

Minor points

In Line 83 the authors already discard many molecules “(lipids, metabolites, etc.)” because they constitute only a small part of the cell’s dry mass, but then later also the contribution of DNA is discarded in Line 93. It is unclear to me why this has to be done in two steps, I would say that either DNA matters or it doesn’t (as a mass constituent).

ρ_{mem} in Figure S2 is called σ_{mem} in Main text.

In Equation 11 of the Appendix, I think μ should be replaced by θ , otherwise I don’t understand how the function $f(y|\theta)$ depends on θ .

In the discussion, it would be interesting to highlight more what the authors envision as a mechanism for controlling the densities of cell structures. In particular, it would be interesting to comment on how these mechanisms are expected to work in individual cells, especially during the cell cycle where surface-to-volume ratio Sa/V and aspect ratio l/w are expected to systematically change while the density of cell constituents is expected to remain constant.

(Remarks on code availability)

Code all seems available and easily runnable.

Version 1:

Reviewer comments:

Reviewer #1

(Remarks to the Author)

Overall assessment

In the revised work, while the main idea and model details remain largely the same, the authors redid all the experiments by using quantitative mass spectrometry. The data quality seems much improved compared to the cell compartmentation assay used in the original work. I still have some concerns as follows. And I hope the authors can address them before further

consideration of the publication in Nature Communications.

Major concerns

1. I agree with reviewer #2's concern that correlation does not mean causation. In the updated work, the ppGpp perturbation is still the sole attempt to test the main prediction, S/V vs. ϕ_{rib} , described by Eq. 4. Since it is the sole perturbation test, the authors should show the thorough data of how exactly each of the variables in Eq. 4 changes at different ppGpp levels. Those variables should include ψ_{mem} , ψ_{peri} , ψ_{cyto} , ρ_{cyto} , ρ_{peri} , and σ_{mem} , since they can be directly measured from the mass spec and cell size images. However, these data are not available in the current manuscript. I had a similar comment in my original report, major issue 3. But the authors responded, "Knowing their exact dependence on ϕ_{rib} becomes necessary only when attempting to predict SA/V across a continuum of ϕ_{rib} values, which is not essential for testing the core principles of our model." In the ppGpp results, ϕ_{rib} does seem to continuously change at different ppGpp levels, so I think the authors should still show those variables' dependence on ϕ_{rib} , and possibly the growth rate. The point is that, if some of the molecular densities, such as ρ_{cyto} or σ_{mem} , are fundamentally controlled properties, they might still be robust in the presence of ppGpp perturbation. This kind of observation itself will provide a more mechanistic insight into what the actual physical constraints are. For example, maybe σ_{mem} is still roughly constant even when κ drops at low ppGpp levels, as the authors showed in Fig. S5B.
2. Related to the first concern, I do not know why the authors did not calculate κ using directly measured ρ_{cyto} and σ_{mem} for the ppGpp experiment. Instead, as shown in Methods 4.9, Eq. 8, they indirectly calculated κ using the mass fractions. The authors should be able to calculate ρ_{cyto} and σ_{mem} from the mass spec and cell size measurement. Can the authors clarify why they chose not to do that?
3. The authors should process literature data to estimate the mass fraction quantities, ψ_{mem} , ψ_{peri} , ψ_{cyto} , and molecular densities, ρ_{cyto} , ρ_{peri} , and σ_{mem} , as they partially did in the original manuscript. And they should compare the literature-based analysis with their own results, check if any of the quantities show general invariance or trends, and discuss them. Those results can be shown as supplemental figures. Again, this kind of cross-comparison could give us some insights on which quantities could be universally regulated, independent of the choice of experimental methods.

Minor issues

1. Lines 64-65: it is ambiguous to say transporters are across all compartment. They are only on the membranes.
2. Fig. 1D caption: the unit of the slope should be hr^{+1} instead of hr^{-1} , since the growth rate has the unit of hr^{-1} .
3. Lines 171-173: the sentence "Specifically ..." is somewhat repetitive to the previous sentence.
4. Fig. 4F caption: "Thin and tick error bars" should be "Thin and thick error bars."

(Remarks on code availability)

The code is well documented and should be ready for further testing by the community.

Reviewer #2

(Remarks to the Author)

Overall comments

This reads like an entirely new paper. The authors have collected an impressive amount of new data that refines their previous work. The data revealing the relationship between growth rate and proteome allocation between the cytoplasm, periplasm, and the membrane is very interesting and valuable, as are the observed relationships. The observations here are important and the correlations are solid and well-established. One valuable insight is that compartments also have limited space and fixed macromolecular densities, and therefore proteome allocation issues. This is an important constraint on how cells work and should be much more clearly stated as it is a solid contribution (and to my knowledge a novel contribution, or at least the experimental data supporting this notion are novel – see my comment on including a reference to Norris et al below). Furthermore, the authors do provide an important phenomenological correlation between cell composition and cell size that to my knowledge has not been established before. It is much more predictive of cell size than growth rate (as the authors explain), and indeed, "growth rate" is probably pretty hard for the cell to measure directly. In general I agree with statements made in the final paragraph of the Discussion.

These correlations are solid, the predictive equation (4) is novel to my knowledge, and new data are valuable. However, the correlations provided do not support many of the claims of new regulatory schemes that are either explicitly made or implied by the authors. In general, the authors often conflate "regulation" with "constraints" and correlation does not indicate causation. The danger here is that something that happens is proposed to arise from some complex regulation scheme, when in reality no added layer of regulation is needed. In the Rebuttal the authors claim to have modified their language to avoid claims of direct causation. This may be so, but the authors have not modified their language enough. Many terms are used imprecisely that will certainly lead non-rigorous readers (which are the majority of the readership of any scientific paper) to imagine regulatory schemes for which the authors have not provided evidence. If the authors are making claims for regulatory schemes, the evidence provided is not nearly strong enough.

Before I list the specific criticisms, I want to mention that this is a very hard problem because cell composition, growth rate, and cell size are strongly correlating properties that are difficult to unravel.

First dubious claim: To take one example, I take serious issue with claims made in Section 2.1: the authors state "proteome allocation is subject to global regulation beyond functional constraints" and "localization dynamics are an integral part of cellular proteome regulation rather than a passive consequence of metabolic demand." I find this doubtful. Proteins go to the compartment where they are needed, and where proteins are needed depends upon the function of the proteins. The statement that "since metabolic proteins – including enzymes and transporter components are distributed across all compartments [Fig. S3] their allocation must be coordinated not just according to function but also spatially within the cell"

needs clarification. Their use of an incredibly broad ontological category (“metabolism”) obscures the point I think they are trying to make. Of course proteins that deal with “metabolism” are distributed across compartments. But I do not see “global regulation.”

The data are certainly supportive of a model in which the different compartments have space limitations, which establishes a constraint on how the metabolic proteins can be allocated between the membrane, the periplasm, and the cytoplasm. In other words, just as different components of the whole-cell proteome need to be allocated properly between functions to optimize growth rate, cellular compartments also have limited proteome space that needs allocation. For instance, there are only so many nutrient transporters that can be added to a membrane due to all the other essential things that membranes do, with the meaningful consequence that surface area density of the membrane might be a limiting factor for the metabolic sector. But the authors should not call this “global regulation.” The size of my apartment does not “globally regulate” how much furniture I have – it constrains how much furniture I have. This is why a better statement would be: “proteome allocation is subject to global constraints established by macromolecular density, in addition to functional constraints.” Invoking “global regulation” (or any regulation) is totally unnecessary. It is indeed interesting to learn how the cell decides to allocate one particular set of proteins with a common function between the compartments – for instance, the xylose utilization operon has membrane channels, periplasmic binding proteins, and cytosolic proteins that convert xylose into central carbon metabolites. Deciding how much the cell makes of each when xylose is available is certainly constrained by packing density in each compartment and by allocation in general (and by enzymatic rates and properties), but this constraint cannot be called “global regulation.”

Second dubious claim: Another highly dubious claim (made in the title as well, and elaborated further in the Discussion) is that the cell maintains a fixed ratio of cytoplasmic macromolecule density to membrane protein density. The idea here is that the cell is somehow monitoring this ratio and is orchestrating things like cell size or width accordingly. This is a far more complicated explanation than is necessary given the data. Specifically, in Figure 2A and 2B, the authors indicate that the densities of proteins in the membrane and cytoplasm are constant across conditions (in their words: “moderately stable”). Because these two densities are constant, the ratio of the cytoplasm to membrane densities is also constant. That’s indeed also true. However, the interpretation that this ratio is fixed by some process that actively couples the density of the cytoplasm with the density of the membrane (sensed e.g. by the Rod complex) is a big overinterpretation. The authors’ claim that the densities of the cytoplasm and the membrane are tightly coupled by one process balancing them both would be much more believable if the two densities were not constant across growth rates and a strict ratio was nevertheless always maintained.

Because the two densities are maintained at a constant level, a simpler and much more plausible explanation can be provided. (Incidentally, the authors state on p. 3 that densities are “moderately stable” while on p. 4 they have a “slight linear dependence” – which one is it?). Occam’s Razor suggests that the density values in the two compartments are independently fixed at constant values for some reason (maybe due to a biophysical constraint or by active regulation of osmolarity). As a consequence, the cytoplasm/membrane density ratio might appear to be coupled and actively maintained at a constant value, when in reality that coupling arises accidentally because two separate phenomena are stabilizing the two densities independently of each other. As I said before, the fact that these two densities are stable is an important finding and very interesting, but there is no evidence supporting a further claim of a coupling mechanism. I am not convinced that the claim made in the manuscript title is substantiated.

Finally, since this claim of density coupling is not supported by the data, it also changes the focus in the Discussion about “how cells sense densities.” Instead of speculating about how the density ratio could be sensed and maintained by e.g. Rod (as there is no evidence that the ratio is maintained in the first place), the authors could speculate about why macromolecular densities in the two compartments are separately maintained at constant levels. This is interesting and could be either active regulation or just some passive consequence of a biophysical property.

Third dubious claim: The claim in the Discussion (p. 7) that a density constraint arises from “simultaneous control over the size of each cellular compartment and the partitioning of the proteome between them” has two issues. First, while the apparent density values appear solid enough for the authors to claim they are constraints, I do not see how the constraints “arise” from “simultaneous control.” Second, this statement seems to suggest there are two regulatory knobs: one knob that makes compartments bigger, and one knob that changes proteome composition. A simpler explanation is that proteome composition (specifically the number of ribosome abundance) is the ONLY regulatory knob, and what looks like “simultaneous control” arises without any further regulation as follows:

- 1) The cell makes more ribosomes,
- 2) Ribosomes necessarily live in the cytoplasmic compartment;
- 3) macromolecular density is constant (so the cell cannot increase cytoplasmic packing);
- 4) therefore, the cytoplasmic compartment must get bigger.

This is not regulating where proteins go, nor does it require regulation of compartment size. Instead, it’s deciding that the cell can afford more ribosomes because metabolic conditions allow it. A straightforward, non-regulation-related consequence of making more ribosomes is that the cytoplasmic compartment gets bigger. No need to separately decide to make a bigger cytoplasm. I would instead state “compartment size changes passively as a byproduct of proteome allocation” unless the authors have evidence to suggest otherwise.

Related to this point, while I agree that the authors have convincingly shown is “macromolecular densities within the cytoplasm [and the membrane] are maintained within a narrow range” I don’t see how this constraint “emerges” from “simultaneous control over the size of each cellular compartment.” This seems backward. Because cells cannot increase macromolecular density, if they want to make more ribosomes, the size of the cytoplasmic compartment must (passively) increase.

Finally, language used in the discussion can easily be taken to mean that cell size is actively regulated through direct control over compartment partitioning. This is confusing to the reader – the image I get is a sorting demon that thinks about cell size and thus sends proteins to either the periplasm or the membrane, when in reality, proteins go where they are

needed and where they can function. Cell size may change as a consequence of these functional requirements, as well as density constraints identified here by the authors. Thus it is more accurate to say “cell size may be a consequence of constraints imposed by compartment partitioning.”

Further points:

1) I also do not understand the argument that “cell composition rather than bulk growth rate is a major determining factor of cell size control.” My problem is the word “determining” – it sounds like the authors are claiming that cell composition is a parameter monitored by some regulatory system, which then sets the cell size. Maybe this is the case but the authors have not demonstrated it here. What the authors have convincingly shown is that “cell composition is a major parameter that is more strongly predictive of cell size than growth rate” which is a very important result.

2) Comment on page 3: “a fundamental trade-off in how cells structure their proteomes; any increase in cytoplasmic protein load is offset by a decrease in periplasmic protein load...” This interesting idea was theoretically explored extensively in Norris et al., “Mechanistic model of nutrient uptake explains dichotomy between marine oligotrophic and copiotrophic bacteria” doi.org/10.1371/journal.pcbi.1009023 – in brief, Norris et al. used simple models to show that oligotrophic bacteria have big periplasms to facilitate nutrient uptake with ABC transporter systems (which use high-affinity periplasmic binding proteins) while copiotrophic bacteria have small periplasms (they do not need ABC transporters), which allows them to afford big cytosols with lots of ribosomes. As this prior work explored these concepts very extensively and related growth rate to compartmental allocation (and tangentially, cell size and compartment size), at minimum the authors must cite this previous work!

(Remarks on code availability)

Reviewer #3

(Remarks to the Author)

Dear editor,

We were delighted to see the improvements that the authors have made to their manuscript. These improvements included serious experimental efforts and an almost complete rewriting of the paper. Since the authors now rely almost solely on their own observations, and no longer on the observations deduced from literature, many of our concerns have been resolved. Up to some minor revisions, we think the manuscript is ready for publication.

Minor comments:

1. In Section 2.1, the authors claim twice that the found scaling relationships for the proteome fractions suggest that there is an active regulation of the localization dynamics:

“This suggests that proteome localization is subject to global regulation beyond functional constraints.”

“This implies that localization dynamics are an integral part of cellular proteome regulation, rather than a passive consequence of demand.”

We find these claims too strong based on the presented data, and also unnecessary for supporting the main results of the paper. As far as we see, the authors do not show evidence that these proteome fractions are scaled because of regulation. In fact, if ψ_{mem} is just kept constant because of a physical constraint, the results of Figure 1 (D),(E),(F) would immediately follow.

2. Relating to the previous point: the information of panels D, E, and F in Figure 1 are mathematically redundant, so we find it somewhat misleading to show these as independent plots. Indeed, when one constrains the sum of three fractions to be 1, and one of these fractions is constant (Fig 1D), the others will perfectly anti-correlate. When two fractions perfectly anti-correlate, their sum is indeed constant. We suggest that the authors pick only the panel they find most important, or that they, at least, make very clear that these panels are not 3 independent pieces of evidence.

3. The claim

“This observation highlights a fundamental trade-off in how cells structure their proteomes; any increase in cytoplasmic protein load is offset by a decrease in periplasmic protein load while the membrane protein partitioning remains unchanged.” is too strong. That two variables anti-correlate in the conditions that were tested doesn’t mean that their sum is “fundamentally” constrained.

4. In Figures 1 and 2, linear fits are shown but the slopes are not reported. Although the authors write which lines are supposedly “linear” and which are “constant”, this is hard to see from the plots. Just reporting the slopes will also not allow for a relevant comparison between different quantities because they are on vastly different scales, so we suggest reporting the slope normalized by the average value for the measured quantity.

5. We think the ppGpp perturbation experiment is key, and can be explained better. The authors perform an orthogonal perturbation to the usual carbon source variation, and they pick it such that the usual relation between ribosome fraction and

growth rate is broken. This is thus a direct test whether the observed SA/V's could also just be described by the often-described relation between growth rate and volume. We think the authors could emphasize this strong choice of experiment more.

In addition, we think it would be good to also show a measured-vs-predicted plot (like in Figure 4E, and F) in Fig S5, which should thus be based on the commonly-assumed relation between volume and growth rate. At least, the authors could mention in which perturbations the growth rate turns out to be a really bad predictor of SA/V.

6. The sentence: "Given a rod-shaped bacterium, this SA/V corresponds to a cell width $w \approx 0.5 \mu\text{m}$, in line with the reported observed minimum width of *E. coli* [Fig. S2(E)]."

seems somewhat misleading. The sentence suggests that the width of 0.5 micrometers is reported in some reference in which they specifically tried to make *E. coli* go thinner. In contrast, Fig S2(F), shows measured cell widths in a collection of datasets. That *E. coli* never becomes thinner than 0.5 micron in these experiments, does not mean that it can not become thinner if necessary.

Note that the reference to Fig S2(E) should be S2(F).

7. Caption of Figure 4: intracellualr -> intracellular

(Remarks on code availability)

Version 2:

Reviewer comments:

Reviewer #1

(Remarks to the Author)

The revised manuscript is good enough for publication if the authors agree to make the following edits.

1. The authors should incorporate their reasoning in response to my second comment into the final version of the main text. By calculating kappa as described in the manuscript, rather than from direct measurements of total protein and RNA mass as I suggested, the authors should acknowledge the limitations of the current methods and provide suggestions for potential future improvements in experimental measurements. This addition can be made either in the section introducing kappa (Eq. 3) or in the Discussion section.
2. The authors should cite Fig. S5 in the main text, either when introducing Fig. 1 or in the Discussion section.

(Remarks on code availability)

The code seems to provide sufficient guidelines for any interested reader to reproduce the data analysis. It also provides other datasets from the literature used in this work.

Reviewer #2

(Remarks to the Author)

This is an impressive data set and a nice analysis that highlights interesting and important correlations in proteome allocation, and the authors should be congratulated for their hard work and solid and insightful contributions. This revised manuscript has addressed my previous comments. The text has been appropriately modified to remove unsupported claims about global regulation. For the Discussion, while I would not suggest that any active sensing by Rod is at play here, the text now reads more like speculation (appropriately so).

(Remarks on code availability)

Reviewer #3

(Remarks to the Author)

All my comments have been addressed by the authors. I would judge this manuscript ready for publication.

(Remarks on code availability)

Response to Reviewers for NCOMMS-24-04046: "Stringent Control Over Cytoplasmic and Membrane Densities Defines Cell Geometry in *Escherichia coli*"

Griffin Chure^{1,*}, Roshali de Silva^{1,2}, Richa Sharma¹, Michael C. Lanz^{1,3}, and Jonas Cremer^{1,*}

Department of Biology, Stanford University, Stanford, CA, USA

1 Department of Biology, Stanford University, Stanford, CA, USA

2 Current Address: School of Life Sciences, Arizona State University, Tempe, AZ, USA

3 Chan-Zuckerberg Biohub, San Francisco, CA, USA

* GC: griffinchure@gmail.com; JC: jonas.cremer@stanford.edu

Author Response: Executive Summary

We thank the reviewers for their very constructive feedback, which has helped us strengthen the work significantly. In response to their comments, we have recollected nearly all datasets used in the work, now using mass spectrometry heavily to test our hypotheses. Given these new data and the new insights they have generated, we have completely rewritten the manuscript and have reduced our reliance on Bayesian inference to probe our theory of density maintenance. Furthermore, this approach eliminates our previous reliance on inference from literature studies, allowing us to directly measure key parameters such as proteome partitioning across cellular compartments. Our new methodology enables more concrete, falsifiable predictions of our density maintenance model and provides a robust test of our hypothesis through expanded ppGpp perturbation experiments. While our core conclusions remain unchanged from the original manuscript, we can now state them with greater certainty and quantitative rigor, having established clear boundaries where our model applies and where it breaks down.

We have also refined our presentation by focusing on our internally consistent dataset, relaxing previously rigid assumptions about parameter constancy, and clearly distinguishing between measured and inferred quantities throughout the manuscript. The revised paper more precisely articulates the relationship between RNA-to-protein ratio and surface-to-volume ratio while avoiding overgeneralization. These improvements have enhanced the manuscript's scientific foundation and clarity, providing a stronger framework for understanding how cell composition relates to bacterial physiology and morphology.

Reviewer #1

Reviewer Comment

This work aims to explain how the RNA-to-protein ratio controls the surface-to-volume ratio of *E. coli* as a consequence of constant protein densities in the cytoplasm and on the membranes. It is an important open question to understand how cellular-level features, such as cell geometry and growth rate, emerge from the microscopic properties, such as the macromolecular composition. The authors revisited more than 30 published datasets, collected new data, and found that they generally support the main idea of this work. I agree with the authors that the protein densities could serve as a fundamental physical constraint to give rise to some general relationships, as discussed in this work. I appreciate the authors' efforts to pool the published datasets. However, I do have a few major concerns listed below, and I think the authors should address them for further consideration of the publication.

1. The authors should discuss in more detail the generality/universality of their theory, formulated as equation 5. This relation seems to hold better across different nutrient conditions, as the authors tried to demonstrate. Specifically, the authors tried to show that RNA/protein ratio is the main control variable, if not the sole one, of the surface-to-volume ratio. However, my question is, is this theory also applicable to a broader range of conditions that *E. coli* experience in both lab and natural conditions? For example,

when the surface-to-volume ratio is significantly perturbed by antimicrobial compounds (Nonejuie et al., PNAS 2013) or genetics tools (Zheng et al., PNAS 2016), does the change in RNA/protein ratio still follow equation 5? Similarly, for equation 7, which links the cell width to RNA/protein, some of the perturbations mentioned above could largely affect the cell width, so do they also alter the RNA/protein accordingly? If not, the authors need to clearly point out that their theory applies only to a limited range of conditions.

Author Response

We appreciate the reviewer's substantive comments and thank them for their careful reading of our manuscript. We agree with many of their concerns and have made significant revisions to address them.

We have substantially revised our approach and manuscript to address the universality/applicability of our theory by basing our analysis entirely on our own experimentally collected data, rather than inferring parameters from previously published studies. This methodological change accomplishes several important improvements:

1. It ensures internal consistency across all measurements, eliminating potential variability from different experimental conditions across datasets.
2. It allows us to directly measure crucial parameters such as proteome partitioning (ψ_{mem} , ψ_{cyto} , ψ_{peri}) rather than inferring them from the literature.

We have also clarified the scope and limitations of our framework. Our model applies primarily to steady-state conditions where proteome partitioning remains within typical physiological variation, such as those observed across different nutrient conditions and ppGpp perturbations. We have carefully adjusted our language throughout the manuscript to more accurately reflect these boundaries.

It is important to note that our study specifically focuses on rod-shaped bacteria, where the surface-to-volume ratio remains approximately constant throughout the cell cycle. This is in contrast to cocci or other morphologies where S_A/V becomes a cell-cycle dependent quantity. While density maintenance principles may be important in these other organisms, the mathematical relationships would differ. We have expanded this point in our discussion section.

Regarding the specific studies mentioned by the reviewer (Nonejuie et al., PNAS 2013; Zheng et al., PNAS 2016), we agree that antimicrobial compounds or direct manipulations of cell division proteins likely introduce complex effects that may not be easily interpreted within our model's framework. In contrast, our approach uses a single targeted perturbation (ppGpp titration) that directly alters a specific parameter of our model (ϕ_{rib}), allowing us to test the resulting effects on cell geometry in a more controlled manner. We believe this provides a more direct and physiologically relevant test of our model's predictions, as it follows a natural regulatory pathway rather than artificially disrupting multiple cellular processes simultaneously. Our revised manuscript explicitly outlines the conditions we have tested and where we believe our model holds true. This careful approach to perturbation was a major motivation for our decision to recollect all experimental data using mass spectrometry, which allowed us to directly measure critical parameters across consistent experimental conditions.

Finally, we have refined our focus to exclusively describe the relationship between RNA-to-protein ratio and surface-to-volume ratio, without extending our model to predict cell width or volume. This more targeted approach prevents overgeneralization beyond conditions where our framework is well-supported by data. We believe these revisions collectively strengthen the scientific foundation of our work while acknowledging its appropriate scope and limitations.

Reviewer Comment

2. I am not very convinced by some of the fitting results. For example, the linear fitting in Fig. 2B

does not seem very good, as the data scatters from ~ 0.05 to ~ 0.18 , almost varying by three folds. The authors need to show the goodness of fit and discuss the likelihood that ϕ_{mem} stays constant across the conditions. Similar issues apply to other fittings, such as Figs. 1C&1D, Figs. 2C&2D, and Figs. 4A-4C.

Author Response

We thank the reviewer for this important critique regarding the quality of our fits. We have taken substantial steps to address these concerns. We have completely recollected all data presented in the paper using mass spectrometry, eliminating our previous reliance on literature studies with potentially inconsistent methodologies. This approach has significantly improved the internal consistency of our dataset and simplified our analysis.

We have relaxed the assumption that ϕ_{mem} (now termed ψ_{mem} in the revised manuscript) remains constant across conditions. In our updated model, we now only assume that it varies linearly with the ribosomal proteome allocation ϕ_{rib} , with a very shallow slope that better captures the observed variation. This modification makes our findings independent of a strict constant membrane proteome fraction assumption and better aligns with our experimental observations.

A key improvement in our revised manuscript is the dramatic reduction in reliance on statistical inference. By directly measuring key parameters through mass spectrometry on a per-replicate basis, we now work with primary data rather than inferred quantities. This approach strengthens the reliability of our results and reduces the uncertainty associated with our previous methodology, which had relied more heavily on Bayesian inference across disparate datasets.

Our revised approach provides a much more direct test of the relationships proposed in our model, and we believe the reviewer will find these improvements address their concerns about the fitting results presented in the original manuscript.

Reviewer Comment

3. In the ppGpp test, to make the argument that S/V follows a similar relation against RNA/protein as shown in Fig. 3D, the authors need to show that ϕ_{mem} and ϕ_{peri} also correlate with RNA/protein in a similar manner to that in Figs. 2B & 2C.

Author Response

We thank the reviewer for this insightful comment regarding the ppGpp test. In our revised manuscript, we have substantially improved our approach to address this concern. Our new mass spectrometry measurements now allow us to directly measure ψ_{mem} and ψ_{peri} across all conditions, including the ppGpp perturbations. This eliminates the need to infer or assume how these quantities scale with ϕ_{rib} when testing our predictions, providing a much more direct and robust assessment of our model.

We have enhanced our analysis by presenting a predicted-versus-measured plot for the surface-to-volume ratio (S_A/V) specifically for the ppGpp perturbations, and have added a new growth condition to strengthen this analysis. This approach allows us to rigorously assess how well our model performs under these perturbations without relying on indirect inferences of ψ_{mem} and ψ_{peri} from ϕ_{rib} measurements.

It is worth noting that the precise functional dependence of ψ_{mem} and ψ_{peri} on ϕ_{rib} is not fundamental to the density maintenance theory itself. The theory requires only that these parameters maintain certain relationships under different conditions. Knowing their exact dependence on ϕ_{rib} becomes necessary only when attempting to predict S_A/V across a continuum of ϕ_{rib} values, which is not essential for testing the core principles of our model.

Together, these improvements in our methodology and analysis provide a more direct test of whether

the relationship between S_A/V and RNA/protein ratio remains consistent under ppGpp perturbations, addressing the reviewer's concern while strengthening the evidence for our model.

Reviewer Comment

While the authors included a lot of published datasets, it is hard to distinguish which datasets are used in different plots. For example, the legends of the figures in Figs. 1A and 1B are mixed. It will be helpful for readers to assess the results if the legends can be separated for different plots.

Another question is related to Fig. 2D, which uses much fewer published datasets in the plot. In principle, because the datasets in Figs. 1A and 1B all measured the growth rates, S/V can be re-calculated from the volume values in Fig. 1B, and plotted against RNA/protein values that fall within a similar growth rate range. This could double-check the robustness of equation 5, although different datasets might use different strain backgrounds, media, and other lab conditions.

Author Response

We thank the reviewer for this helpful suggestion regarding the clarity of our figure legends. In our revised manuscript, we have made a substantial change that directly addresses this concern: we no longer include any literature data in the main text of the work. All of our main figures now exclusively feature our newly collected data, which provides a consistent and coherent presentation throughout the paper.

The only place where we now compare our data to literature values is in Supplementary Figure S2. In this supplementary figure, we have carefully implemented the reviewer's suggestion by ensuring clear identification of each dataset, including placing distinct legends adjacent to their respective panels. This organization makes it much easier for readers to identify which specific datasets are being presented in each plot. This change not only improves the clarity of our presentation but, in our view, also strengthens our scientific argument by focusing on a consistent and internally controlled dataset in the main text.

Reviewer Comment

Minor Issues:

1. Can the authors justify the use of Bayesian inferential model and the need of probability distributions of RNA/protein? It seems that simple means such as standard errors is sufficient to evaluate the uncertainties of the measurements.

Author Response

We appreciate the reviewer's suggestion to cross-validate our findings by re-calculating S/V from volume values across datasets. In our revised manuscript, we have taken a more fundamental approach to address concerns about robustness by completely recollecting all data using consistent methodology.

Our new approach presents measurements on a per-replicate basis, with experimental uncertainties that are often smaller than the data points themselves on the plots. This comprehensive recollection of data has eliminated the need for much of the inference we performed in the original manuscript and provides a more direct test of equation 5. Rather than attempting to reconcile potentially inconsistent datasets from the literature (which, as the reviewer correctly notes, come from different strain backgrounds, media, and laboratory conditions), we now rely entirely on our own internally consistent measurements. This approach avoids the complications that would arise from trying to normalize across

heterogeneous published datasets.

We still employ Bayesian inferential models to estimate different quantities and to propagate uncertainty in our theoretical predictions. We believe that using a Bayesian definition of probability allows us to explicitly and quantitatively define our *a priori* assumptions about what range of values these parameters could plausibly take. Our methods and appendix sections now include a brief justification of our use of Bayesian probability. This revised approach provides a more rigorous and direct assessment of the relationships in our model than would be possible through recalculation across heterogeneous literature datasets.

Reviewer Comment

2. Typo in line 51: “Wiving.” 3. Typo in line 15: “Escherichia. coli” should be “Escherichia coli.”

Author Response

We’ve now corrected these typos and thank the reviewer for their close read of the manuscript.

Reviewer # 2

Reviewer Comment

The authors have compiled an impressive set of data from publications spanning decades. As far as I am aware, the authors are the first to integrate proteomics data sets together with cell size measurements in order to demonstrate that membrane protein density and cytoplasmic dry mass density are tightly constrained in *E. coli*. This is an interesting and important finding on its own.

Next, the authors claim that given these constraints, variations in ribosomal content must give rise to variations in cell size. The authors claim that this explains correlations between growth rate, cell dimensions, and ribosomal content. This is an interesting and controversial claim, and an intriguing alternative to existing hypotheses. Whether this is a consequence of a system that coordinates composition and cell size (as the authors propose), or is instead simply a correlation arising from the cell regulating other unrelated processes (a spandrel, defined in Wikipedia as “a phenotypic characteristic that is a byproduct of the evolution of some other characteristic, rather than a direct product of adaptive selection” – also see ref 24 in the MS) has not been clearly established by the authors.

The authors have convincingly demonstrated correlations; however, concluding that correlations must reflect insightful phenomenological growth laws is not straightforward because it is important to distinguish two possibilities: a) there is causality behind the correlation, or b) the observed correlation is simply a spandrel (a null interpretation). An example of a phenomenological finding that has provided real biological insight is the work by Scott et al (Science 2010, reference 9 in the MS). Scott et al. use a correlation between RNA/protein ratio and growth rate to establish the theory of proteome allocation. Importantly, Scott et al. challenged their own interpretation in various ways: by artificially inflating ribosome numbers with chloramphenicol, and by overexpressing a useless protein to alter proteome allocation. These tests were essential in helping to establish that resource allocation is indeed an essential principle underlying the correlation between RNA/protein ratio and growth rate. For the present study, such tests are lacking. Is there an experiment that can disprove the proposed hypothesis? Such a test that can help distinguish whether the authors are demonstrating a meaningful regulatory link between RNA/protein ratio and cell size.

For instance, one possible “null interpretation” would be that dry mass per cell volume is maintained by a system that monitors osmolarity (e.g., membrane channels). Furthermore, this system could easily be imagined as highly robust and completely indifferent to cell geometry, growth rate, and the

RNA/protein ratio. This strikes me as a much simpler explanation. In the absence of an independent test of the hypothesis that demonstrates a direct link, it remains possible that the correlation observed is not a growth law but actually a consequence of a very robust macromolecular density homeostasis system that coincidentally occurs together with a system that couples cell size with (for instance) DNA replication timing. In that case, the authors' proposal of a system that specifically couples cell size to density would be incorrect.

On the other hand, if cell density homeostasis indeed constrains cell geometry via an indirect mechanism, I would argue that the authors have actually discovered that cell geometry is an emergent property of cell density being rigorously controlled by osmolarity maintenance systems acting together with the systems that maintain cell shape (e.g. the Rod complex). Discovering that cell size is an indirect consequence of density regulation would be very interesting and an important contribution!

So in summary, I have two significant conceptual criticisms: 1) The authors have a hypothesis inspired by a correlation, but do not propose or perform an experiment that has a chance of disproving this hypothesis. There is a danger that the hypothesis is actually untestable.

2) The authors do not address the possibility that drymass density might be regulated independently of cell geometry. If the authors can demonstrate that cell size is a spandrel resulting from robust density maintenance, that would nicely address this criticism!

Author Response

We thank the reviewer for their thoughtful assessment of our work. We appreciate the recognition of our efforts to integrate proteomics datasets with cell size measurements, demonstrating that membrane protein density and cytoplasmic dry mass density are tightly constrained in *E. coli*. Regarding the reviewer's conceptual criticisms, we have made several important revisions to address these concerns:

First, we have conducted additional experiments that specifically test the predictive breaking of growth rate dependence on size and ribosomal content. By examining perturbations of ppGpp concentrations in two conditions (glucose and glucose+CAA), we pushed our system into regimes where our model predictions fail, serving as a critical test of the boundaries where this relationship holds true. Importantly, we also provide a plausible explanation for why the model breaks in these specific regimes. This addresses the reviewer's concern about the testability of our hypothesis. We acknowledge that our work tests only one type of perturbation (ppGpp titration), and in our discussion, we now propose a specific candidate mechanism involving RodZ that may regulate this relationship. While detailed investigation of this mechanism lies outside the scope of this work—similar to how the mechanisms underlying the resource allocation principles in Scott *et al.* were elucidated in subsequent studies by Dai and others—we believe our identification of this relationship and its limitations represents an important contribution.

In response to the second criticism, we've refined our analysis to specifically consider cytoplasmic macromolecular density rather than total dry mass density. This distinction is crucial, as cytoplasmic macromolecular density is intrinsically connected to geometry in ways that make independent regulation difficult to reconcile with physical constraints. Our revised data and analysis more clearly illustrate this connection. We've also modified our language throughout the manuscript to avoid implying direct causation while emphasizing that these correlations are remarkably strong and mutually predictive across both conditions and perturbations in ways that have not been previously demonstrated. We believe these revisions address the reviewer's concerns while maintaining the significance of our findings: that cell geometry appears to be an emergent property arising from the interplay between density maintenance and proteome allocation. Whether this relationship represents a directly regulated system or an indirect consequence of other cellular constraints, the predictive power of our model provides valuable insight into bacterial cell physiology.

Reviewer Comment

Other suggestions: The authors rightly point out that cell size and growth rate can be readily decoupled (as seen in Buke et al (ref 70) and Basan et al (ref 21)), as can growth rate and ribosomal abundance (as seen in Scott et al., and many other publications). The authors propose instead that cell composition is a “major determining factor of cell size control.” The authors provide some support for this notion by modulating ppGpp and observing that the RNA/protein ratio and S/V ratio change in a manner predicted by their theory (Figure 3). However, it looks like the authors are using only 5 data points. As mentioned above, many similar experiments have been performed in which RNA/protein ratio is decoupled from growth rate and cell size: chloramphenicol titrations in Scott et al., and overexpression of “useless” proteins, which enlarges cells while slowing growth (Scott et al. and Basan et al.) Do results from such decoupling experiments also agree with the density maintenance theory? If so, the authors should show this explicitly in figures: e.g. one figure of cell size vs. growth rate that indicates decoupling after expressing useless proteins or after chloramphenicol treatment and then showing that even in these cases, the relationship between cell composition and cell size remains solid. If those literature data from decoupling conditions are already present in Figure 1, the authors should unpack such data as it would better support the proposal that cell composition is a “major determining factor of cell size control” even when growth rate is decoupled from cell size and cellular composition. This would be a much more convincing illustration than the 5 data points from the RelA/Mesh1 experiments in Figure 3.

Author Response

We appreciate the reviewer’s thoughtful suggestions regarding additional support for our density maintenance theory through decoupling experiments. We have substantially strengthened our approach in response to these concerns. Our revised manuscript now relies entirely on a new dataset where we directly measured all model parameters except for the density ratio (which we determine through fitting). This approach eliminates our previous dependence on a collection of literature studies with potentially inconsistent methodologies and provides a much more robust test of our hypothesis.

Regarding the ppGpp perturbations specifically, we have significantly expanded this analysis, quadrupling the number of measurements from 5 to 20 data points. This expansion allows us to conduct a much more rigorous comparison between predicted and measured values. Additionally, we have introduced a new supplemental figure (Fig. S5) which demonstrates that S_A/V can be decoupled from the growth rate via Mesh1 overexpression while the density ratio remains conserved. We believe this addresses the reviewer’s concern about the limited data points in our previous analysis.

We agree with the reviewer that examining other perturbations such as chloramphenicol treatment or overexpression of “useless” proteins would be valuable. However, properly testing whether these perturbations conform to the density maintenance hypothesis requires simultaneous measurement of both cell size and proteome composition/partitioning. Unfortunately, this complete set of measurements cannot be extracted from the existing literature, as the studies cited by the reviewer did not measure proteome partitioning between cellular compartments.

Our ppGpp perturbations are particularly well-suited for testing our hypothesis because they have more targeted effects than chloramphenicol treatment, which can cause complex physiological responses beyond altering ribosome content. The expanded dataset for these perturbations now provides a more convincing demonstration of how cell composition relates to cell size even when growth rate is partially decoupled from these parameters.

We have also revised our language throughout the manuscript to more precisely characterize the relationship between cell composition and size control, avoiding overstatement while still highlighting the predictive power of our model across the conditions we have tested.

Reviewer Comment

Page 2: “Wiving matter” (should be “Living matter”)

Page 2: “As there are enumerable interfacial interactions” – do the authors mean “innumerable interfacial interactions”?

Author Response

These errors have now been corrected, and we thank the reviewer for their close read of the manuscript.

Reviewer # 3

Reviewer Comment

I’m almost sure that some of the posterior probability intervals that are being reported are not the relevant ones. For example, the authors note in line 69 that “both densities are remarkably tightly constrained with median values of $\rho_{cyt} = 287.09^{+5.26}_{-5.21}$ fg / fL and $\sigma_{mem} = 2.7^{+0.4}_{-0.3}$ fg / μm^2 [Fig. 1(E)] where the sub- and super-scripts denote the lower and upper bounds of the 95% credible regions”

This could indeed be, exactly as the authors write, the 95% posterior prob. interval on the median density. However, when the authors want to claim that the densities are tightly constrained it is not enough to show that the posterior on the median is tightly constrained, as this is mostly a reflection of the number of datapoints. Instead, it would be better to simply report the variance on the measured densities, which seems to be much larger than the reported error-bars.

This is even clearer in line 106, where the authors claim “First, we observe that the membrane proteome fraction is a fixed quantity at $\phi_{mem} = 0.131^{+0.006}_{-0.006}$ (blue lines), suggesting that while the expression of individual membrane components may vary across conditions, the total membrane protein fraction is fixed.” Given the shown datapoints, it seems quite a ridiculous claim that the fraction is a “fixed quantity” within a range of 0.012, while the figure clearly shows datapoints ranging from 0.05 to 0.15, and while most datapoints fall out of the reported 95% posterior prob. interval. Again, it could indeed be that the reported range is a reasonable uncertainty region on the estimated average fraction, but this does not say that the membrane proteome fraction is fixed.

I suggest that a relevant measure of the variation is reported, and that the claims are re-worded to clearly state that there is still significant variation, even though the quantities seem somewhat stable across conditions and datasets.

Author Response

We thank the reviewer for their careful and close reading of the manuscript, particularly regarding our presentation of uncertainty and claims about “fixed” parameters. Their points are well-taken, and we have made substantial changes to address these concerns in our revised manuscript. Most importantly, we have completely changed both our dataset and our inference procedure. Our analysis now relies exclusively on our new measurements rather than inferred parameters from literature data. This approach eliminates many of the statistical concerns raised by the reviewer, as we no longer strictly rely on posterior estimates for dry mass density or membrane density where the uncertainty intervals could be misinterpreted.

We have also revised our language throughout the manuscript to more accurately reflect the observed variation in these parameters. We have relaxed our previous assumption that quantities like membrane proteome fraction must be constant across conditions. Instead, where we have assumed the density ratio is constant, we have done so explicitly and fit our model to the data with this assumption

in place, yielding posterior credible regions that properly capture the full variance of the data (see new Fig. 3(D)).

These changes provide a more accurate representation of the statistical evidence supporting our model while avoiding overstatement about the degree of parameter constancy. We believe this revised approach addresses the reviewer's concerns while maintaining the scientific value of our findings.

Reviewer Comment

Relating to this, maybe the authors can provide plots in which the data of Figures 2B, 2C is stratified per dataset as a supplemental figure. To me, it seems that the two newest (and maybe most reliable datasets), i.e. from the authors themselves and from Mori et al. (2021), show a negative trend of the membrane proteome fraction with the ribosomal proteome fraction. In the authors' dataset, only the last point breaks this negative trend. However, it is not easy to judge this from this figure because the datasets are all overlaid, and the plot is therefore quite crowded. All in all, I'm not too convinced by the authors' claims that ϕ_{mem} is a fixed quantity, but I also think this is not strictly necessary for the authors' main conclusion to hold.

I suggest to remove the data-points of older studies in Figure 2D, and keep only the datapoints that have been gathered by the authors themselves. It would of course be good to re-use data if there was sufficient information present, however, if I understand it correctly, these datasets only report surface-to-volume ratios and growth rates. It is quite a stretch to test the authors' theory with that data. Because it requires the reader to accept that:

1. ϕ_{mem} is well-described by the constant blue line in Figure 2B. The blue line doesn't even describe the data very well on which it was fitted, let alone other data. So this is quite a stretch.
2. ϕ_{peri} is well-described by the purple line in Figure 2C. Even though this fit seems a bit more convincing, remember that the data used in Figure 2D is not shown in this plot because there is no proteome information for these older datasets.
3. The empirical relation between growth rate and RNA/prot-ratio describes the data in these older datasets very well. Although this relation is the most established, it is clear from Figure 1A that there is plenty of variation left around the fitted line.

All in all, I find it impossible to assume all this and see this as independent proof of the authors' theory. Since the datapoints from these older datasets are based on SA/V and growth rate only, the plotted points are in fact just a replotting of the known relation between SA/V and growth rate. Although it is in principle interesting that the authors can reproduce this relation using their theory, it should not be presented as proof of their more complicated theory.

Author Response

We thank the reviewer for their insightful analysis regarding the limitations of our original approach. We agree with many of their points, particularly regarding the difficulty in using older datasets with limited measurements to test our model rigorously.

In our revised manuscript, we have addressed these concerns through several fundamental changes. Most significantly, we now rely exclusively on our new dataset for the main analysis, rather than inferring parameters from previously published studies. This new dataset includes direct mass spectrometry measurements of the ppGpp perturbations, allowing us to make a direct comparison between the predicted S_A/V under the density maintenance model and the measured S_A/V . These experiments, coupled with our revised inference approach for the density ratio κ , serve as a much more robust test of our hypothesis.

As the reviewer correctly noted, our model does not necessarily depend on an assumption that the membrane partition ψ_{mem} (termed ϕ_{mem} in the previous draft) is constant. We have now relaxed this assumption, allowing this parameter to vary as a function of the ribosomal proteome allocation ϕ_{rib} under a phenomenological linear model with a shallow slope. We appreciate the reviewer's suggestion to relax this assumption and believe this has substantially improved our analysis.

Importantly, our expanded experiments have revealed a regime where our hypothesis no longer holds. We provide a compelling rationale for why this occurs, highlighting the boundaries of our model's applicability. This finding strengthens rather than weakens our overall conclusions by establishing clear limits to the density maintenance hypothesis.

Regarding the reviewer's suggestion to remove or stratify literature data, we now use the literature data only as a supplementary comparison (Fig. S2) to check the consistency of our observations and ensure we are not overfitting to phenomena exclusive to our dataset. In this supplementary figure, we have improved the presentation so that each literature study is more clearly identifiable.

These revisions collectively address the reviewer's concerns about overinterpretation while providing a more rigorous test of our central hypothesis.

Reviewer Comment

I would suggest that the authors also add a scatterplot to Figure 2 of measured S_A/V versus predicted S_A/V according to their theory for their data, where they use the data from 2B and 2C directly (i.e. without using the fitted relations). Based on the figures, it seems that these predicted and measured values would match fairly nicely. In fact, I think this would make the theory a bit more general because it would show that Equation 2A holds independent of whether the membrane-proteome fraction is constant or not. Since the authors claim Figure 2 mostly "strongly support[s] a hypothesis that density maintenance defines the cellular surface-to-volume," this also should not depend on the behavior of the membrane proteome fraction.

The ppGpp-perturbation experiments are a very important contribution of this paper because they directly test the proposed theory. I do think Figure 3 can be made quite a bit stronger and more informative. First, why is the theoretical prediction (i.e. the green line) from Figure 2D not shown in Figure 3D?

Author Response

This suggestion by the reviewer was a major motivation to undertake the new suite of experiments to directly measure proteome partitioning in all of our data. We think this is a major and substantive improvement to the work, and we thank the reviewer for suggesting it.

In the original manuscript, we could not put the green line (i.e. our prediction of S_A/V vs ϕ_{rib}) over our ppGpp measurements as we did not know how the ppGpp perturbations would alter the proteome partitioning parameters ψ_{mem} and ψ_{peri} . Our new dataset directly measures these parameters and allows us to directly calculate the predicted S_A/V as a function of composition and partitioning given our estimate of κ . As we fit κ from the wildtype data, it does not make much sense to plot a predicted versus measured S_A/V for the wildtype as a confirmation of our model. However, we now show the wildtype predicted-versus-measured in the new figure Fig. 4(E & F) to give a sense of the noise in our estimate. The fact that the ppGpp perturbations largely fall within the noise of the wildtype measurements illustrates that our model does a very good job predicting the S_A/V for the ppGpp perturbations.

Reviewer Comment

Second, although it is informative to see the posterior probability distributions in 3B and 3C I would advise also showing the actual datapoints. Currently, it is unclear how many datapoints these posteriors are based on, and it can also not be checked quickly by the reader if the posterior distributions look reasonable. I see that there are some data-lines shown in the Supplement, but since these are marked "our experimental measurements" I can't tell which relate to Figure 2 and which to Figure 3

Author Response

We agree with the reviewer and we now show the per-replicate measurements for these quantities in the revised Fig. 4.

Reviewer Comment

Third, and most importantly, if the authors want to show that the growth rate does not set the SA/V-ratio in these experiments, the reader needs to be provided with this information. However, no information on the growth rate is present, and we can thus not decide whether the relation based on a dependence of SA/V on growth rate is wrong, even though the authors claim to disprove this theory when they say: "In summary, these findings strongly supports the claim that cell geometry is set by the cell composition, and not the details of the particular growth condition." So this information on the growth rate needs to be added, ideally in a plot that shows that the new datapoints don't fall on the empirical growth rate vs SA/V-line.

Author Response

We thank the reviewer for this important point regarding the need to show growth rate information to substantiate our claim about cell geometry being set by composition rather than growth conditions. We have addressed this directly in our revised manuscript.

We have added a new supplementary figure (Fig. S5) that explicitly plots SA/V versus growth rate for all perturbations. This figure clearly demonstrates that the Mesh1 perturbation deviates from the "wildtype" relationship line, providing direct evidence that S_A/V can be decoupled from growth rate under certain conditions. Interestingly, the RelA perturbations still largely follow the wildtype trend line, which is consistent with our understanding of how these different perturbations affect cellular physiology.

Additionally, our revised Fig. 4 now includes a specific panel showing the direct effect of ppGpp perturbation on growth rate across both tested growth conditions, alongside the effects on other key parameters. This new panel in Fig. 4 reveals a non-monotonic change in growth rate with induction of either Mesh1 or RelA, while demonstrating monotonic changes in both ribosomal allocation (ϕ_{rib}) and surface-to-volume ratio (S_A/V). This non-alignment of trends provides clear evidence that these quantities can be decoupled from growth rate.

Importantly, despite this decoupling from growth rate, our prediction of S_A/V under the density maintenance model remains accurate. This observation directly supports our claim that cell geometry is primarily determined by cellular composition rather than by growth rate itself. We believe these additions to the manuscript provide the necessary information for readers to evaluate this central claim of our work.

Reviewer Comment

The points brought up in the discussion by the authors on temperature affecting growth rate without affecting cell geometry or composition, as well as the results from Basan et al. showing that changing RNA/prot by overexpressing a useless cytosolic protein affects cell size while the density is maintained should be brought up earlier. Showing these results and comparing them to their prediction if data is available or measuring it themselves would be a great support to their claim.

Author Response

We agree with the reviewer that having these data would improve our claims. However, performing these experiments along with the necessary mass spectrometry measurements they would require, lies outside the scope of the current work. We believe that the revised text of the manuscript now addresses these points well.

Reviewer Comment

In Figure 4 I am doubting if the RNA/prot ratios that are shown for the old datasets are measured or again inferred as in Figure 2 using the ribosome growth law. In the case of the latter, this should be noted in the text.

This relates to a more general point, which is that it would be better if the authors always state more clearly what information is directly measured, inferred from proteomics, or inferred from empirical relations. I am convinced that the authors' findings are non-trivial and interesting, but reading the current version of the manuscript, one is easily led to believe that everything is circular and trivial.

Author Response

We appreciate the reviewer's concern about clarity regarding measured versus inferred data in our original manuscript. We agree completely that greater transparency was needed, and we have addressed this issue thoroughly in our revised manuscript.

Our new approach dramatically reduces reliance on inferred parameters. The revised manuscript now uses only our direct measurements for all main analyses and conclusions. Specifically:

- All RNA/protein ratios presented in the main figures are now directly measured values from our experiments, not inferred from empirical relations.
- We have clearly labeled all figures and figure captions to distinguish between measured quantities and any inferred parameters.
- We have restructured our presentation to emphasize direct measurements over inferences, with almost all presented findings now being based on primary experimental data.
- In the few instances where inference is still necessary, we explicitly state this in both the main text and figure captions, along with a description of the inference method.

We believe these changes address the reviewer's concern about potential circularity and make the non-trivial nature of our findings much more apparent. The direct measurement approach in our revised manuscript provides a much stronger foundation for our conclusions while maintaining clarity about the source and reliability of all data presented.

Reviewer Comment

Generally, the authors claim in the Abstract that their work “provides a candidate biochemical mechanism for how cell size control is manifested.” I think this promise is not delivered upon, and I would suggest removing this from the Abstract. The theory that the authors propose is quite far from being a biochemical mechanism, as it is not clear how this density is maintained. For example, is the cell actively controlling the densities, or are the densities automatically kept constant because the corresponding area and volumes just expand because new proteins are made and take up space? Relating to that, I don’t think the authors can claim that they have shown “Stringent control ...” in the title. Whether the constant densities are due to stringent control or due to emergent biophysical properties is an interesting question that cannot be claimed solved, but rather should be addressed in the Discussion.

Author Response

We agree with both points raised by the reviewer and our new draft of the manuscript address these comments. Specifically, we have removed all comment on "stringent control" and have softened our language surrounding "mechanism". We now emphasize that our work presents a candidate biophysical quantity that can be plausibly measured by the cell, but do not assert this is a mechanism.

Reviewer Comment

Minor points:

In Line 83 the authors already discard many molecules (“lipids, metabolites, etc.”) because they constitute only a small part of the cell’s dry mass, but then later also the contribution of DNA is discarded in Line 93. It is unclear to me why this has to be done in two steps, I would say that either DNA matters or it doesn’t (as a mass constituent).

Author Response

We agree with the reviewer and this has now been addressed.

Reviewer Comment

ρ_{mem} in Figure S2 is called σ_{mem} in Main text.

Author Response

We thank the reviewer for their close read of the manuscript and have now ensured that all notation is clear and consistent throughout the work.

Reviewer Comment

In Equation 11 of the Appendix, I think mu should be replaced by theta, otherwise I don’t understand how the function $f(y|\theta)$ depends on theta.

Author Response

We've now fixed this issue and streamlined our description of the inference, ensuring notational consistency and clarity.

Reviewer Comment

In the discussion, it would be interesting to highlight more what the authors envision as a mechanism for controlling the densities of cell structures. In particular, it would be interesting to comment on how these mechanisms are expected to work in individual cells, especially during the cell cycle where surface-to-volume ratio S_A/V and aspect ratio ℓ/w are expected to systematically change while the density of cell constituents is expected to remain constant.

Author Response

This is an excellent point and we thank the reviewer for their suggestion, which we have now incorporated into the discussion. In our revised manuscript, we have expanded our discussion of potential regulatory mechanisms, with specific attention to the Rod complex as a central player in this process. A key insight we emphasize in our revision is that for rod-shaped bacteria like *E. coli*, the surface-to-volume ratio (S_A/V) does not change appreciably throughout the cell cycle. As illustrated in Fig. 5(A), while the cell length changes substantially as the cell grows, the width remains approximately constant within a given growth condition. This geometric property means that $S_A/V \approx 4/w$ when the length is significantly greater than the width, providing a straightforward mechanism by which cells can control S_A/V solely by tuning their width.

We now propose a more detailed hypothesis regarding how density homeostasis might be maintained at the single-cell level. Specifically, we suggest that the Rod complex, which rotates through both the cytoplasmic and membrane environments during cell wall synthesis, may serve as both an effector and sensor of the relative density between these compartments. As the Rod complex experiences density-dependent forces during its rotation, these mechanical cues could modulate its activity to ensure coordination between length increase and width control.

In our revised discussion, we have been more explicit about the speculative nature of this mechanism while providing a plausible biophysical framework that connects proteome composition and localization to cell geometry. We note that genetic perturbations of various Rod complex components strongly affect cell size and shape homeostasis, supporting their potential role as sensors of relative density between membrane and cytoplasm.

This proposed feedback mechanism provides a conceptual framework for how cells might maintain relatively constant densities despite the dynamic changes occurring throughout the cell cycle, bridging the gap between molecular-level composition and organism-level geometry.

Response to Reviewers for NCOMMS-24-04046: "Maintenance of Cytoplasmic and Membrane Densities Shapes Cellular Geometry in *Escherichia coli*"

Griffin Chure^{1,*}, Roshali de Silva^{1,2}, Richa Sharma¹, Michael C. Lanz^{1,3}, and Jonas Cremer^{1,*}

Department of Biology, Stanford University, Stanford, CA, USA

1 Department of Biology, Stanford University, Stanford, CA, USA

2 Current Address: School of Life Sciences, Arizona State University, Tempe, AZ, USA

3 Chan-Zuckerberg Biohub, San Francisco, CA, USA

* GC: griffinchure@gmail.com; JC: jbremer@stanford.edu

Author Response: Executive Summary

We thank the reviewers for their continued engagement and constructive feedback on our revised manuscript. In response to their comments, we have made targeted revisions to improve language and interpretation. We have, for example, adapted reviewer-suggested phrasing that frames our observations in terms of biophysical constraints rather than active regulatory processes. We have clarified that our core contribution is a quantitative, predictive framework (Eq. 4) linking cellular composition to geometry, regardless of the specific underlying mechanisms. We have also addressed technical concerns by adding several new supplementary figures, including comparisons with growth rate-based predictions that demonstrate the superior predictive power of our composition-based model. We believe these revisions strengthen the manuscript and address all of the reviewers' concerns.

Reviewer #1

Reviewer Comment

Overall assessment

In the revised work, while the main idea and model details remain largely the same, the authors redid all the experiments by using quantitative mass spectrometry. The data quality seems much improved compared to the cell compartmentation assay used in the original work. I still have some concerns as follows. And I hope the authors can address them before further consideration of the publication in Nature Communications.

Major concerns

1. I agree with reviewer #2's concern that correlation does not mean causation. In the updated work, the ppGpp perturbation is still the sole attempt to test the main prediction, S_A/V vs. ϕ_{rib} , described by Eq. 4. Since it is the sole perturbation test, the authors should show the thorough data of how exactly each of the variables in Eq. 4 changes at different ppGpp levels. Those variables should include ψ_{mem} , ψ_{peri} , ψ_{cyto} , ρ_{cyto} , ρ_{peri} , and σ_{mem} , since they can be directly measured from the mass spec and cell size images. However, these data are not available in the current manuscript. I had a similar comment in my original report, major issue 3. But the authors responded, "Knowing their exact dependence on ϕ_{rib} becomes necessary only when attempting to predict S_A/V across a continuum of ϕ_{rib} values, which is not essential for testing the core principles of our model." In the ppGpp results, ϕ_{rib} does seem to continuously change at different ppGpp levels, so I think the authors should still show those variables' dependence on ϕ_{rib} , and possibly the growth rate. The point is that, if some of the molecular densities, such as ρ_{cyto} or σ_{mem} , are fundamentally controlled properties, they might still be robust in the presence of ppGpp perturbation. This kind of observation itself will provide a more mechanistic insight into what the actual physical constraints are. For example, maybe σ_{mem} is still roughly constant even when κ drops at low ppGpp levels, as the authors showed in Fig. S5B.

Author Response

We appreciate the reviewer's close read of the manuscript and are grateful that they agree the data quality of the work is now much improved. We also appreciate their suggested new figures and data presentation. We've now included this new figure as Fig S6, with the caveat discussed below. With our data set, we are able to plot how the partitioning coefficients ψ_{cyto} , ψ_{peri} , and ψ_{mem} change under ppGpp perturbation as a function of ϕ_{rib} , S_A/V , and the growth rate λ . However, we are unable to compute the plot how the densities ρ_{cyto} , ρ_{peri} , and σ_{mem} change as function of these variables as they are not directly measured, but are inferred from concomitant total protein measurements. Unfortunately, we were unable to reliably make total protein measurements alongside our growth curves, mass spectrometry, and cell size imaging on a per-sample basis, and thus cannot compute these quantities without making unreasonable assumptions, as described in the next response.

Reviewer Comment

2. Related to the first concern, I do not know why the authors did not calculate κ using directly measured ρ_{cyto} and σ_{mem} for the ppGpp experiment. Instead, as shown in Methods 4.9, Eq. 8, they indirectly calculated kappa using the mass fractions. The authors should be able to calculate ρ_{cyto} and σ_{mem} from the mass spec and cell size measurement. Can the authors clarify why they chose not to do that?

Author Response

We agree that this point requires clarification. We have accordingly updated our discussion in the second revision.

In short, the direct calculation of ρ_{cyto} and σ_{mem} from mass spectrometry and cell size measurements would require additional measurements that were not experimentally feasible for our ppGpp experiments.

A direct calculation of cytoplasmic density would require knowledge of total protein mass $M_{prot}^{(tot)}$, total RNA mass $M_{RNA}^{(tot)}$, the cytoplasmic partitioning coefficient ψ_{cyto} , and cell volume V_{tot} for each ppGpp induction condition,

$$\rho_{cyto}(ppGpp) = \frac{\psi_{cyto}(ppGpp)M_{prot}^{(tot)}(ppGpp) + M_{RNA}^{(tot)}(ppGpp)}{V_{tot}(ppGpp)}. \quad (R1)$$

Similarly, the membrane protein areal density would require total protein content, the membrane protein partitioning coefficient ψ_{mem} , and cellular surface area S_A for each induction condition,

$$\sigma_{mem}(ppGpp) = \frac{\psi_{mem}(ppGpp)M_{prot}^{(tot)}(ppGpp)}{2S_A(ppGpp)}. \quad (R2)$$

While we measured proteomic partitioning and mass fractions, cell sizes, and bulk growth rates, unfortunately measuring $M_{prot}^{(tot)}$ and $M_{RNA}^{(tot)}$ for each biological replicate in each growth condition was not experimentally feasible. These quantities could be inferred by assuming how they scale with inducer concentration, but we lack compelling *a priori* knowledge to make such approximations.

However, we realized that by taking the ratio of these densities and making a very reasonable assumption that the RNA-to-protein ratio is proportional to ribosomal proteome allocation ϕ_{rib} , κ can be

calculated using only the quantities we could reliably measure:

$$\kappa(ppGpp) = \frac{2(\psi_{cyto} + \beta\phi_{rib}(ppGpp))}{S_A(ppGpp)\psi_{mem}(ppGpp)} \quad (R3)$$

This approach avoids the need for separate total mass measurements while still providing a rigorous calculation of κ . We have clarified this reasoning in both the main text and appendix.

Reviewer Comment

3. The authors should process literature data to estimate the mass fraction quantities, ψ_{mem} , ψ_{peri} , ψ_{cyto} , and molecular densities, ρ_{cyto} , ρ_{peri} , and σ_{mem} , as they partially did in the original manuscript. And they should compare the literature-based analysis with their own results, check if any of the quantities show general invariance or trends, and discuss them. Those results can be shown as supplemental figures. Again, this kind of cross-comparison could give us some insights on which quantities could be universally regulated, independent of the choice of experimental methods.

Author Response

We appreciate the reviewer's suggestion to add a figure comparing our measurements of the partition coefficients ψ_{cyto} , ψ_{mem} , and ψ_{peri} and densities ρ_{cyto} , σ_{mem} , and ρ_{peri} . We have now included this analysis as an additional supplementary figure (now Fig. S5 in the revised manuscript).

We note that in order to compute the densities, we rely on phenomenological fits of the total protein, total RNA, cell volume, and cell surface area as a function of the growth rate, as there is no other data set which measures all of these quantities simultaneously for the same strain in the same growth condition.

Nevertheless, this new figure shows that the partitioning coefficients from our mass spectrometry measurements are in very good agreement with literature measurements. A notable exception is our measurement of the periplasmic proteome partition ψ_{peri} , for which we routinely observe a larger partition.

We note that there is also disagreement in our estimates of the compartment macromolecular densities, with all densities being approximately two-fold higher than those inferred from a combination of literature measurements. However, despite this disagreement, our measurements and those from the literature (with the exception of Valgepea et al. 2015) show an approximately constant value for the cytoplasm-membrane density ratio with quantitatively comparable values.

Overall, we believe that the dataset we've collected for this manuscript is the most robust and self-consistent measurement of all of these quantities on a per-replicate and per-condition basis significantly reducing the amount of inference needed to test our hypotheses of density maintenance and cell geometry control. Performing a full integration of our new data set with literature studies would require significantly more inference and uncertainty quantification (a point that was critiqued in our initial submission), which we believe is outside the scope of our current analysis.

Reviewer Comment

Minor issues

1. Lines 64-65: it is ambiguous to say transporters are across all compartment. They are only on the membranes.

Author Response

We have adjusted the phrasing of this sentence in the revised draft. We now note that, while transporters are exclusively in the membrane, there are frequently soluble periplasmic and cytoplasmic components (such as in ABC transport systems) which contribute to densities across all compartments.

Reviewer Comment

2. Fig. 1D caption: the unit of the slope should be hr instead of hr^{-1} , since the growth rate has the unit of hr^{-1} .

Author Response

We apologize for this error. We have now corrected this mistake in all figures where slopes are reported.

Reviewer Comment

3. Lines 171-173: the sentence “Specifically ...” is somewhat repetitive to the previous sentence.
4. Fig. 4F caption: “Thin and tick error bars” should be “Thin and thick error bars.”

Author Response

These errors have been remedied and we thank the reviewer for their attention to detail.

Reviewer 2

Reviewer Comment

This reads like an entirely new paper. The authors have collected an impressive amount of new data that refines their previous work. The data revealing the relationship between growth rate and proteome allocation between the cytoplasm, periplasm, and the membrane is very interesting and valuable, as are the observed relationships. The observations here are important and the correlations are solid and well-established. One valuable insight is that compartments also have limited space and fixed macromolecular densities, and therefore proteome allocation issues. This is an important constraint on how cells work and should be much more clearly stated as it is a solid contribution (and to my knowledge a novel contribution, or at least the experimental data supporting this notion are novel – see my comment on including a reference to Norris et al below). Furthermore, the authors do provide an important phenomenological correlation between cell composition and cell size that to my knowledge has not been established before. It is much more predictive of cell size than growth rate (as the authors explain), and indeed, “growth rate” is probably pretty hard for the cell to measure directly. In general I agree with statements made in the final paragraph of the Discussion.

These correlations are solid, the predictive equation (4) is novel to my knowledge, and new data are valuable. However, the correlations provided do not support many of the claims of new regulatory schemes that are either explicitly made or implied by the authors. In general, the authors often conflate “regulation” with “constraints” and correlation does not indicate causation. The danger here is that something that happens is proposed to arise from some complex regulation scheme, when in reality no added layer of regulation is needed. In the Rebuttal the authors claim to have modified their language

to avoid claims of direct causation. This may be so, but the authors have not modified their language enough. Many terms are used imprecisely that will certainly lead non-rigorous readers (which are the majority of the readership of any scientific paper) to imagine regulatory schemes for which the authors have not provided evidence. If the authors are making claims for regulatory schemes, the evidence provided is not nearly strong enough. Before I list the specific criticisms, I want to mention that this is a very hard problem because cell composition, growth rate, and cell size are strongly correlating properties that are difficult to unravel.

Author Response

We appreciate Reviewer 2's recognition that this represents 'an entirely new paper' with an 'impressive amount of new data' and that our correlations are 'solid' and our predictive equation is 'novel.' We also value their acknowledgment that this work represents 'an important phenomenological correlation'.

We further want to emphasize that we agree that phrasing is very important when reporting our partitioning and density relations, to prevent confusion and wrong interpretations as much as possible. We particularly recognize in this context the reviewer's preference for framing findings as constraints rather than coordinated 'regulatory' processes. We have adjusted our language accordingly when presenting and commenting on our findings. This includes softening mechanistic claims and emphasizing the phenomenological nature of our findings. We hope this will assuage the reviewer's concerns. In the following, we address the different points by the reviewer in more detail.

Reviewer Comment

First dubious claim: To take one example, I take serious issue with claims made in Section 2.1: the authors state "proteome allocation is subject to global regulation beyond functional constraints" and "localization dynamics are an integral part of cellular proteome regulation rather than a passive consequence of metabolic demand." I find this doubtful. Proteins go to the compartment where they are needed, and where proteins are needed depends upon the function of the proteins. The statement that "since metabolic proteins – including enzymes and transporter components are distributed across all compartments [Fig. S3] their allocation must be coordinated not just according to function but also spatially within the cell" needs clarification. Their use of an incredibly broad ontological category ("metabolism") obscures the point I think they are trying to make. Of course proteins that deal with "metabolism" are distributed across compartments. But I do not see "global regulation." The data are certainly supportive of a model in which the different compartments have space limitations, which establishes a constraint on how the metabolic proteins can be allocated between the membrane, the periplasm, and the cytoplasm. In other words, just as different components of the whole-cell proteome need to be allocated properly between functions to optimize growth rate, cellular compartments also have limited proteome space that needs allocation. For instance, there are only so many nutrient transporters that can be added to a membrane due to all the other essential things that membranes do, with the meaningful consequence that surface area density of the membrane might be a limiting factor for the metabolic sector. But the authors should not call this "global regulation." The size of my apartment does not "globally regulate" how much furniture I have – it constrains how much furniture I have. This is why a better statement would be: "proteome allocation is subject to global constraints established by macromolecular density, in addition to functional constraints." Invoking "global regulation" (or any regulation) is totally unnecessary. It is indeed interesting to learn how the cell decides to allocate one particular set of proteins with a common function between the compartments – for instance, the xylose utilization operon has membrane channels, periplasmic binding proteins, and cytosolic proteins that convert xylose into central carbon metabolites. Deciding how much the cell makes of each when xylose is available is certainly constrained by packing density in each compartment and by allocation in general (and by enzymatic rates and properties), but this constraint cannot be called "global regulation."

Author Response

We appreciate the reviewer's detailed analysis of our language regarding proteome allocation. While we respectfully disagree that we made dubious claims we agree that it is better to frame our observations in terms of physical constraints rather than regulatory processes as suggested by the reviewer. We rephrased the text accordingly and particularly prevent the use of 'global regulation'. In fact, we realized the text is improved by not commenting too much on any consequences or underlying reasons when reporting the well-defined relations in proteome partitioning across cell compartments. We have shortened the text accordingly.

Reviewer Comment

Second dubious claim: Another highly dubious claim (made in the title as well, and elaborated further in the Discussion) is that the cell maintains a fixed ratio of cytoplasmic macromolecule density to membrane protein density. The idea here is that the cell is somehow monitoring this ratio and is orchestrating things like cell size or width accordingly. This is a far more complicated explanation than is necessary given the data. Specifically, in Figure 2A and 2B, the authors indicate that the densities of proteins in the membrane and cytoplasm are constant across conditions (in their words: "moderately stable"). Because these two densities are constant, the ratio of the cytoplasm to membrane densities is also constant. That's indeed also true. However, the interpretation that this ratio is fixed by some process that actively couples the density of the cytoplasm with the density of the membrane (sensed e.g. by the Rod complex) is a big overinterpretation. The authors' claim that the densities of the cytoplasm and the membrane are tightly coupled by one process balancing them both would be much more believable if the two densities were not constant across growth rates and a strict ratio was nevertheless always maintained. Because the two densities are maintained at a constant level, a simpler and much more plausible explanation can be provided. (Incidentally, the authors state on p. 3 that densities are "moderately stable" while on p. 4 they have a "slight linear dependence" – which one is it?). Occam's Razor suggests that the density values in the two compartments are independently fixed at constant values for some reason (maybe due to a biophysical constraint or by active regulation of osmolarity). As a consequence, the cytoplasm/membrane density ratio might appear to be coupled and actively maintained at a constant value, when in reality that coupling arises accidentally because two separate phenomena are stabilizing the two densities independently of each other. As I said before, the fact that these two densities are stable is an important finding and very interesting, but there is no evidence supporting a further claim of a coupling mechanism. I am not convinced that the claim made in the manuscript title is substantiated.

Finally, since this claim of density coupling is not supported by the data, it also changes the focus in the Discussion about "how cells sense densities." Instead of speculating about how the density ratio could be sensed and maintained by e.g. Rod (as there is no evidence that the ratio is maintained in the first place), the authors could speculate about why macromolecular densities in the two compartments are separately maintained at constant levels. This is interesting and could be either active regulation or just some passive consequence of a biophysical property.

Author Response

We appreciate the reviewer's comments about the ratio of densities and our discussion of potential sensing mechanisms. In response, we have substantially revised the discussion section to address these concerns while preserving what we believe are valuable and justified speculations, appropriate for a discussion part of a paper to encourage discussion and future research. In short, the revised text now addresses the following points:

- We reformulated the text to clarify that the Rod complex represents one possible mechanism

rather than the definitive explanation, using language such as "represents a plausible candidate" and "could potentially be subjected to density-dependent forces"

- We added appropriate caveats by stating upfront that "the specific mechanisms underlying density maintenance remain unclear"
- The text now emphasizes the robustness of our framework by concluding that "Irrespective of the molecular mechanisms at play, our predictive framework of density maintenance provides a quantitative foundation for understanding these relationships and their physiological consequences"

We further removed 'coordination' from the title to clarify that we demonstrate the maintenance of densities without implying mechanisms about their implementation. We believe that these changes maintain the scientific value of discussing potential mechanisms while making clear that our core findings—the quantitative relationship described by Eq. 4—stand independently of any specific mechanistic interpretation.

Reviewer Comment

Third dubious claim: The claim in the Discussion (p. 7) that a density constraint arises from "simultaneous control over the size of each cellular compartment and the partitioning of the proteome between them" has two issues. First, while the apparent density values appear solid enough for the authors to claim they are constraints, I do not see how the constraints "arise" from "simultaneous control." Second, this statement seems to suggest there are two regulatory knobs: one knob that makes compartments bigger, and one knob that changes proteome composition. A simpler explanation is that proteome composition (specifically the number of ribosome abundance) is the ONLY regulatory knob, and what looks like "simultaneous control" arises without any further regulation as follows:

- 1) The cell makes more ribosomes,
- 2) Ribosomes necessarily live in the cytoplasmic compartment;
- 3) macromolecular density is constant (so the cell cannot increase cytoplasmic packing);
- 4) therefore, the cytoplasmic compartment must get bigger.

This is not regulating where proteins go, nor does it require regulation of compartment size. Instead, it's deciding that the cell can afford more ribosomes because metabolic conditions allow it. A straightforward, non-regulation-related consequence of making more ribosomes is that the cytoplasmic compartment gets bigger. No need to separately decide to make a bigger cytoplasm. I would instead state "compartment size changes passively as a byproduct of proteome allocation" unless the authors have evidence to suggest otherwise. Related to this point, while I agree that the authors have convincingly shown is "macromolecular densities within the cytoplasm [and the membrane] are maintained within a narrow range" I don't see how this constraint "emerges" from "simultaneous control over the size of each cellular compartment." This seems backward. Because cells cannot increase macromolecular density, if they want to make more ribosomes, the size of the cytoplasmic compartment must (passively) increase. Finally, language used in the discussion can easily be taken to mean that cell size is actively regulated through direct control over compartment partitioning. This is confusing to the reader – the image I get is a sorting demon that thinks about cell size and thus sends proteins to either the periplasm or the membrane, when in reality, proteins go where they are needed and where they can function. Cell size may change as a consequence of these functional requirements, as well as density constraints identified here by the authors. Thus it is more accurate to say "cell size may be a consequence of constraints imposed by compartment partitioning."

Author Response

We share the notion how cell-size follows changes in composition described by the reviewer. We also

appreciate the reviewer's suggested language. To prevent confusion and improve the text we have incorporated specific phrasing suggestions. Particularly, we now write: "Macromolecular densities within the cytoplasm and the areal density of proteins on the cell membrane are maintained within a narrow range. Cell geometry may change as a byproduct of proteome partitioning to maintain these densities". We have also removed language that could be misinterpreted as implying active coordination of cell size through protein targeting decisions.

We recognize this is largely a discussion about how to interpret the same empirical observations, and future work has to investigate to which extent changes in size are passive or actively regulated. Importantly, our core contribution - the quantitative, predictive relationship between cellular composition and geometry described by Eq. 4 - remains unchanged regardless of whether the underlying mechanisms involve active coordination or passive consequences of biophysical constraints. This mathematical framework provides predictive power independent of mechanistic interpretation. We believe our revised language better reflects the distinction between observed mathematical relationships and inferred regulatory mechanisms.

Reviewer Comment

Further points:

1) I also do not understand the argument that "cell composition rather than bulk growth rate is a major determining factor of cell size control." My problem is the word "determining" – it sounds like the authors are claiming that cell composition is a parameter monitored by some regulatory system, which then sets the cell size. Maybe this is the case but the authors have not demonstrated it here. What the authors have convincingly shown is that "cell composition is a major parameter that is more strongly predictive of cell size than growth rate" which is a very important result.

2) Comment on page 3: "a fundamental trade-off in how cells structure their proteomes; any increase in cytoplasmic protein load is offset by a decrease in periplasmic protein load..." This interesting idea was theoretically explored extensively in Norris et al., "Mechanistic model of nutrient uptake explains dichotomy between marine oligotrophic and copiotrophic bacteria" doi.org/10.1371/journal.pcbi.1009023 – in brief, Norris et al. used simple models to show that oligotrophic bacteria have big periplasms to facilitate nutrient uptake with ABC transporter systems (which use high-affinity periplasmic binding proteins) while copiotrophic bacteria have small periplasms (they do not need ABC transporters), which allows them to afford big cytosols with lots of ribosomes. As this prior work explored these concepts very extensively and related growth rate to compartmental allocation (and tangentially, cell size and compartment size), at minimum the authors must cite this previous work!

Author Response

We appreciate the reviewer's suggested language change and have revised our text accordingly. We now state that "cell composition is a major property that is more strongly predictive of cellular geometry than growth rate" rather than claiming it is a "determining factor." This indeed better reflects what our data demonstrate - predictive relationships rather than regulatory mechanisms.

We thank the reviewer for bringing the Norris et al. work to our attention. We have added this citation and acknowledge their theoretical exploration of the trade-offs between periplasmic and cytoplasmic allocation.

Reviewer 3

Reviewer Comment

We were delighted to see the improvements that the authors have made to their manuscript. These improvements included serious experimental efforts and an almost complete rewriting of the paper. Since the authors now rely almost solely on their own observations, and no longer on the observations deduced from literature, many of our concerns have been resolved. Up to some minor revisions, we think the manuscript is ready for publication.

Author Response

We appreciate that the reviewers concerns about inference from literature data sets have been resolved in the new analysis. We thank them for their original review as it drove us to generate new experimental measurements and we strongly believe the paper has markedly improved as a result.

Reviewer Comment**Minor comments:**

1. In Section 2.1, the authors claim twice that the found scaling relationships for the proteome fractions suggest that there is an active regulation of the localization dynamics:

“This suggests that proteome localization is subject to global regulation beyond functional constraints.” “This implies that localization dynamics are an integral part of cellular proteome regulation, rather than a passive consequence of demand.”

We find these claims too strong based on the presented data, and also unnecessary for supporting the main results of the paper. As far as we see, the authors do not show evidence that these proteome fractions are scaled because of regulation. In fact, if ψ_{mem} is just kept constant because of a physical constraint, the results of Figure 1 (D),(E),(F) would immediately follow.

Author Response

We agree with the reviewer that our language in section 2.1 should be softened and 'regulation' should be avoided as a term. To which extent global regulation is involved and to which extent relations follow due to more passive adjustments needs to be investigated in future studies. While editing the text, we further realized that it is best not to comment too much on any consequences or underlying reasons when first reporting the well-defined relations in proteome partitioning across cell compartments. We have shorted the text accordingly.

Reviewer Comment

2. Relating to the previous point: the information of panels D, E, and F in Figure 1 are mathematically redundant, so we find it somewhat misleading to show these as independent plots. Indeed, when one constrains the sum of three fractions to be 1, and one of these fractions is constant (Fig 1D), the others will perfectly anti-correlate. When two fractions perfectly anti-correlate, their sum is indeed constant. We suggest that the authors pick only the panel they find most important, or that they, at least, make very clear that these panels are not 3 independent pieces of evidence.

Author Response

We appreciate the reviewers concern and apologize that details of this portion of our analysis were not clearly conveyed. Importantly, we are not constraining that the partition of these components is equal to unity. In addition to the three partitions of the cytoplasm, membrane, and periplasm, a number of proteins are associated as "external/secreted" as defined by Babu et al. 2018. While these proteins are very rarely detected in our analysis ($\ll 1\%$), they include components such as the flagellar filament FliC which can constitute a significant protein component in high-motility conditions. Our inclusion of this partition does not force the sum of all three partitions to be equal to one. However, we make this approximation explicitly in the model derivation section. As such, these panels are not three independent pieces of evidence as rightly noted by the reviewer. We have now adjusted the main text to make this point abundantly clear. Specifically, we now write in Section 2.1 of the manuscript:

"Notably, the cytoplasmic and periplasmic partitioning trends are near-perfectly anticorrelated across growth rates [Fig. 1(E)], in line with a constant membrane partition ψ_{mem} . This is further illustrated by a constant total partition of the proteome to the cytoplasm and periplasm ($\psi_{cyto} + \psi_{peri}$) across all growth conditions, accounting for $\sim 87\%$ of the proteome mass [Fig. ??(F)]."

Reviewer Comment

3. The claim "This observation highlights a fundamental trade-off in how cells structure their proteomes; any increase in cytoplasmic protein load is offset by a decrease in periplasmic protein load while the membrane protein partitioning remains unchanged." is too strong. That two variables anti-correlate in the conditions that were tested doesn't mean that their sum is "fundamentally" constrained.

Author Response

We agree with the reviewer. We have removed the term "fundamental" in the revised manuscript and further softened the language. We write: "This observation suggests a trade-off in how cells structure their proteome partitioning: any increase in cytoplasmic protein load is offset by a decrease in periplasmic protein load an vice versa, while the membrane protein partitioning remains unchanged."

Reviewer Comment

4. In Figures 1 and 2, linear fits are shown but the slopes are not reported. Although the authors write which lines are supposedly "linear" and which are "constant", this is hard to see from the plots. Just reporting the slopes will also not allow for a relevant comparison between different quantities because they are on vastly different scales, so we suggest reporting the slope normalized by the average value for the measured quantity.

Author Response

We appreciate this suggestion. While our initial revised manuscript included slopes for Fig. 1 in the figure caption, we now have included a new supplemental table (Table 1) presenting all slopes and intercepts for the linear fits performed in the work as a whole.

Reviewer Comment

5. We think the ppGpp perturbation experiment is key, and can be explained better. The authors perform an orthogonal perturbation to the usual carbon source variation, and they pick it such that the usual relation between ribosome fraction and growth rate is broken. This is thus a direct test whether the observed S_A/V 's could also just be described by the often-described relation between growth rate and volume. We think the authors could emphasize this strong choice of experiment more.

Author Response

We appreciate the reviewer's comment that our perturbation of ppGpp is a critical test, yet is not sufficiently explained. We therefore have adjusted the text of our introduction to this perturbation. The first paragraph of which now reads as follows:

"Our analysis so far suggests that the surface-to-volume ratio S_A/V can be accurately described by a simple model defined by a constant cytoplasm-membrane density ratio κ and a variable ribosomal allocation ϕ_{rib} . This is confirmed from cells grown in different carbon sources with a range of growth rates, and therefore a range of ϕ_{rib} . However, in contrast to other cell size models, cell geometry in our model (Eq. 4) does not explicitly depend on the cellular growth rate or the specifics of the environment, but only on the cellular composition. This provides an opportunity to test our model by composition perturbations.

Here we specifically perturbed the global regulator guanosine tetraphosphate (ppGpp) which break the typical correlation between ribosomal content ϕ_{rib} and growth rate, allowing us to test whether S_A/V follows composition (as Eq. 4 predicts) or growth rate (as conventional models suggest)."

Reviewer Comment

In addition, we think it would be good to also show a measured-vs-predicted plot (like in Figure 4E, and F) in Fig S5, which should thus be based on the commonly-assumed relation between volume and growth rate. At least, the authors could mention in which perturbations the growth rate turns out to be a really bad predictor of S_A/V .

Author Response

We thank the reviewer for this suggestion. We have now included an additional supplemental figure (Fig S7) which displays this analysis. We find that the induction of Mesh1 (increasing ppGpp) incurs a significant change in volume which deviates from the the classical exponential volume growth law. We have also adjusted the text to explicitly discuss this finding and further strengthen our observation that the surface-to-volume is correctly predicted under the density maintenance hypothesis.

Reviewer Comment

The sentence: "Given a rod-shaped bacterium, this S_A/V corresponds to a cell width $w \approx 0.5 \mu\text{m}$, in line with the reported observed minimum width of *E. coli* [Fig. S2(E)]." seems somewhat misleading. The sentence suggests that the width of 0.5 micrometers is reported in some reference in which they specifically tried to make *E. coli* go thinner. In contrast, Fig S2(F), shows measured cell widths in a collection of datasets. That *E. coli* never becomes thinner than 0.5 micron in these experiments, does not mean that it can not become thinner if necessary. Note that the reference to Fig S2(E) should be S2(F).

Author Response

We agree with the reviewer and have revised the phrasing of this section to make it clear that the apparent limit of $\approx 0.5 \mu\text{m}$ is a strictly empirical bound. Additionally, we have fixed the incorrect figure reference.

Reviewer Comment

7. Caption of Figure 4: intracellualr -> intracellular

Author Response

This typo has now been corrected. We thank the reviewer for their close read of the manuscript.

Response to Reviewers for NCOMMS-24-04046: "Maintenance of Cytoplasmic and Membrane Densities Shapes Cellular Geometry in *Escherichia coli*"

Griffin Chure^{1,*}, Roshali de Silva^{1,2}, Richa Sharma¹, Michael C. Lanz^{1,3}, and Jonas Cremer^{1,*}

Department of Biology, Stanford University, Stanford, CA, USA

1 Department of Biology, Stanford University, Stanford, CA, USA

2 Current Address: School of Life Sciences, Arizona State University, Tempe, AZ, USA

3 Chan-Zuckerberg Biohub, San Francisco, CA, USA

* GC: griffinchure@gmail.com; JC: jbremer@stanford.edu

Author Response: Executive Summary

We thank the reviewers for their continued engagement and constructive feedback. We are happy to hear that all three reviewers assess the work ready for publication. We adjusted the manuscript following the remarks of Reviewer 1 (see below). We further adjusted the abstract to fit within the 150 word limit.

Reviewer Comment

Reviewer 1 (Remarks to the Author): The revised manuscript is good enough for publication if the authors agree to make the following edits. 1. The authors should incorporate their reasoning in response to my second comment into the final version of the main text. By calculating kappa as described in the manuscript, rather than from direct measurements of total protein and RNA mass as I suggested, the authors should acknowledge the limitations of the current methods and provide suggestions for potential future improvements in experimental measurements. This addition can be made either in the section introducing kappa (Eq. 3) or in the Discussion section. 2. The authors should cite Fig. S5 in the main text, either when introducing Fig. 1 or in the Discussion section.

Author Response

Following the reviewer's suggestion we now refer to Fig. S5 (with corrected order now Supplementary Figure 4) in the main text where we also introduce more directly the comparative analysis of different datasets from the literature. We also refer to Supplementary Fig. 4 in the captions of Figs. 1 and 2.

We further adjusted the result section to clarify how kappa was inferred. Since this is relevant for the ppGpp perturbation experiments we comment on this in the last part of the result section. We further clarify in the discussion that more direct measurements of densities and density ratios systematically across different genetic perturbations are an important direction for future studies. We thank the reviewer for their thoughtful review and are pleased to hear they judge it ready for publication.

Reviewer Comment

Reviewer 2 (Remarks to the Author): This is an impressive data set and a nice analysis that highlights interesting and important correlations in proteome allocation, and the authors should be congratulated for their hard work and solid and insightful contributions. This revised manuscript has addressed my previous comments. The text has been appropriately modified to remove unsupported claims about global regulation. For the Discussion, while I would not suggest that any active sensing by Rod is at play here, the text now reads more like speculation (appropriately so).

Author Response

We thank the reviewer for their thoughtful review and are pleased that they recommended our work for publication.

Reviewer Comment

Reviewer 3 (Remarks to the Author): All my comments have been addressed by the authors. I would judge this manuscript ready for publication.

Author Response

We thank the reviewer for their thoughtful review and are pleased that they recommended our work for publication.